# Deciphering polymorphism in 61,157 *Escherichia coli* genomes via epistatic sequence landscapes

Lucile Vigué [1,6], Giancarlo Croce[2,3,6], Marie Petitjean [1], Etienne Ruppé[1,4], Olivier Tenaillon [1,7 ✉] & Martin Weigt [5,7 ✉]

Characterizing the effect of mutations is key to understand the evolution of protein sequences and to separate neutral amino-acid changes from deleterious ones. Epistatic interactions between residues can lead to a context dependence of mutation effects. Context dependence constrains the amino-acid changes that can contribute to polymorphism in the short term, and the ones that can accumulate between species in the long term. We use computational approaches to accurately predict the polymorphisms segregating in a panel of 61,157 *Escherichia coli* genomes from the analysis of distant homologues. By comparing a context-aware Direct-Coupling Analysis modelling to a non-epistatic approach, we show that the genetic context strongly constrains the tolerable amino acids in 30% to 50% of amino-acid sites. The study of more distant species suggests the gradual build-up of genetic context over long evolutionary timescales by the accumulation of small epistatic contributions.

[1] Université Paris Cité and Université Sorbonne Paris Nord, Inserm, IAME, F-75018 Paris, France. [2] Department of Oncology, Ludwig Institute for Cancer Research Lausanne, University of Lausanne, Lausanne, Switzerland. [3] Swiss Institute of Bioinformatics—SIB, Lausanne, Switzerland. [4] Laboratoire de Bactériologie, Hôpital Bichat, APHP, Paris, France. [5] Sorbonne Université, CNRS, Institut de Biologie Paris Seine, Computational and Quantitative Biology—LCQB, Paris, France. [6] These authors contributed equally: Lucile Vigué, Giancarlo Croce. [7] These authors jointly supervised this work: Olivier Tenaillon, Martin Weigt. ✉email: olivier.tenaillon@inserm.fr; martin.weigt@sorbonne-universite.fr

Understanding how biological diversity emerges and evolves is at the heart of molecular evolutionary biology. The long-standing confrontation between adaptationists[1] and neutralists[2] has oriented the scientific debate towards comparing the relative contributions of natural selection and drift in the process. While the first ones consider most of the differences between organisms to result from adaptation to different environments, the second one support that polymorphisms reflect mostly random occurrences of equally fit variants.

In recent years, the increasing interest in the role played by historical contingency has revived this old neutral-versus-selective debate[3]. Evolutionary contingency arises when mutations that fix depend on permissive mutations that occurred before. Once fixed, they influence the fate of upcoming mutations and become increasingly deleterious to remove—a phenomenon called entrenchment[4]. The concept of contingency puts epistasis at the forefront of molecular evolution: an amino acid that is neutral or beneficial in a genetic context, can be deleterious in another due to epistatic interactions between residues[5]. Characterizing these epistatic interactions is thus key to uncover the context dependence of mutation effects and understand the extent to which contingency shapes molecular evolution. Moreover, predicting which non-synonymous mutations are likely or not to affect a protein is essential in molecular genetics. Though genetic studies from quantitative trait locus (QTL) analyses to genome-wide association studies (GWAS) successfully identify genomic regions associated to a disease or to a trait of interest, these regions usually encompass multiple neutral mutations in addition to the causative one. An accurate characterization of non-synonymous mutation effects would definitely help identifying the causative mutations.

Deep mutational scans and small adaptive landscape reconstructions allow to experimentally study the effect of mutations or combinations of mutations in a genetic background[3,6]. They highlight the short-term evolutionary constraints the protein faces and a more general pattern of negative epistasis in which deleterious mutations become more deleterious in combination. However, purifying selection removes these mutations from the population. Consequently, their epistatic interactions may not contribute to long-term protein evolution. Some experiments have unveiled a strong role of positive epistasis over long evolutionary times, by measuring the effect of the same mutation in distant homologs from diverged or ancestral species[7,8]. For instance, the same amino-acid change can be deleterious in distant backgrounds while being neutral or beneficial in its native background.

Computational approaches can help to bridge the gap between short-term and long-term evolution. On the one hand, simulations can mimic the fixation of amino-acid changes across many generations[4,9–11]. Yet, their results rely on the validity of the assumptions made to model protein evolution and the effects of epistasis. On the other hand, data-driven approaches to study protein evolution become possible thanks to the revolution of high-throughput sequencing. The accumulation of closely related and more diverged genome sequences enables us to track the emergence and the fixation of amino-acid changes over different timescales. Instead of simulating evolution, we can analyze the patterns of diversity observed in nature on both short-term (polymorphisms within a species) and longer-term (fixed differences between diverged species). The computational study of epistasis requires models of amino-acid sequences that account for epistatic interactions between residues. A current tool to model epistasis is Direct-Coupling Analysis (DCA). DCA is a statistical physics-based approach[12] that aims at modeling the statistical constraints acting on divergent but homologous protein sequences. Indeed, differences between homologous sequences most often represent harmless or, very rarely, beneficial mutations that have been allowed by evolution to persist as they lead to functional proteins. For example, if a residue is conserved throughout the alignment of homologous protein sequences, it is likely crucial to the functionality of the protein and a mutation would produce a large detrimental effect. Similarly, due to epistatic interactions, a pair of amino acids may appear with a different frequency than what would be expected based on conservation of the respective residues. DCA aims to model statistical patterns (e.g., conservation or correlation patterns, see Methods for more details) in protein sequence alignments and relate them to the protein's biological structure and function. It successfully identified residue contacts in the three-dimensional protein fold[12], generated new and functional artificial enzymes[13], predicted deep mutational scanning outcomes[14,15] and was used to investigate amino-acid changes between two closely related genomes[16]. For all these applications, DCA epistatic models consistently perform better than simpler non-epistatic modeling approaches (independent models, IND, often used in bioinformatics for homology detection and sequence alignment). In contrast with other epistasis-aware methods that can be used for predictions[17,18], DCA is explicitly parameterized in terms of epistatic couplings and conservation, making it interpretable.

In this work, we use IND and DCA models in a large-scale study of the *Escherichia coli* core genome in order to understand to what extent epistasis constrains the emergence of non-synonymous polymorphisms within a species, and how these epistatic constraints are building up through time. We do so by first predicting the level of variability at each amino-acid site using both IND and DCA models; subsequently, we confront these predictions to the variability observed across natural *E. coli* isolates. To this end, we have gathered a collection of >60,000 *E. coli* genomes. The analysis is complemented by using a sample of diverged species ranging from *Escherichia coli* to *Yersinia pestis* to study fixed differences accumulating with increasing sequence divergence. With the statistical power of this genome-scale approach, we show: (i) that mutation effect prediction can identify the sites where polymorphisms segregate; (ii) that we can quantify the contribution of the genetic background to these predictions; (iii) that epistatic interactions build up slowly over evolutionary timescales.

## Results

**Data-driven protein sequence landscapes for the case proteome of *E. coli*.** The central concept of our work are amino-acid sequence landscapes, constructed for each protein or protein domain in some reference genome, here *E. coli*. These landscapes associate a DCA score $E$ to any sequence $(a_1, ..., a_L)$. A DCA score is composed of single-residue terms reflecting amino-acid conservation and pairwise couplings modeling epistatic interactions between pairs of residues. Low DCA scores correspond to fully functional sequences whereas high values to non-functional ones (Fig. 1). We build these amino-acid sequence landscapes by training DCA models on multiple-sequence alignments (MSAs) of distant homologs sampled in diverged species (see the sections "Datasets—interspecies MSAs" and "DCA and IND models"). These are widely variable sequences (typical sequence identities are around 20–30%), so they may be understood as a global sample of the sequence landscape, cf. the dark blue dots in Fig. 1. To avoid biasing the results, we have removed from the MSAs any sequence which is too close to *E. coli* (more than 90% identity in sequence). Therefore, it is not evident that the resulting models are informative about the very local structure of the landscape around the *E. coli* reference sequence (white and light blue dots in Fig. 1). The latter might be dominated by idiosyncratic

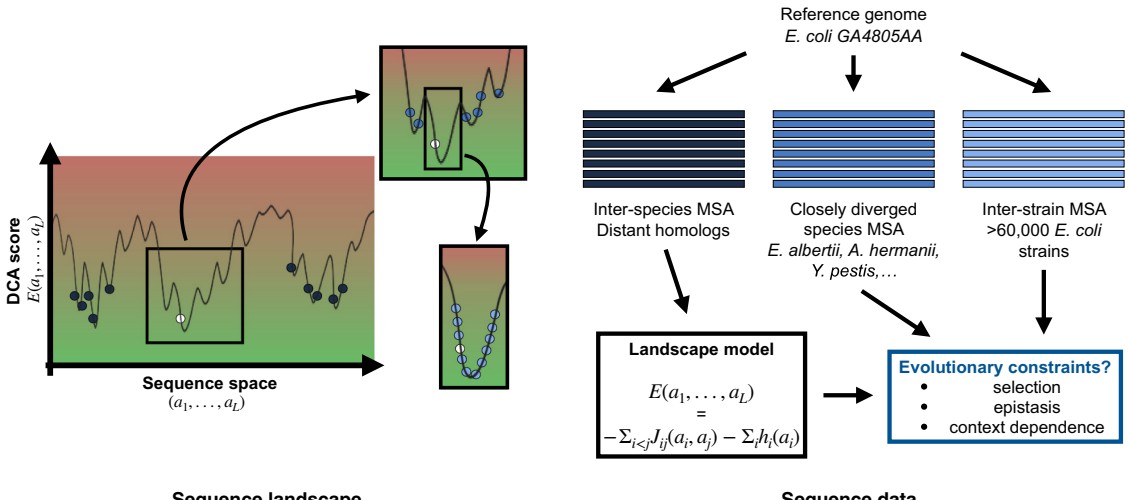

**Fig. 1 Schematic representation of the sequence landscape and its relation to sequence data.** The landscape is defined via a real-valued function of any aligned sequence, with low values indicating "good" functional sequences (green area), and high values "bad" non-functional sequences (red area). Natural sequences can be seen as samples of low values: close orthologs (light blue) of a reference sequence (in white) form a sample which is localized in sequence space and surrounded by closely diverged species (mid-blue). Distantly diverged homologs (dark blue) form a global sample. All sequence data are aligned relative to the reference sequence. Within our work, the global sample will be used to infer data-driven landscape models for all proteins families present in the *E. coli* core genome, and the variability of the local sample and the closely diverged species will be analyzed for signatures of selection, epistasis and context dependence of natural amino-acid polymorphisms.

constraints characterizing *E. coli* as a species, while the MSAs of homologs contain the conserved evolutionary constraints of the entire protein family. Thus, we want to investigate whether amino-acid sequence landscapes can unify the study of epistasis on short and long evolutionary timescales.

**Strong signature of selection at the amino-acid level.** We first test, how accurately DCA can model *E. coli* amino-acid sequences. To work at a genome scale, we focus on 2053 Pfam domains[19] spanning 281,513 residues among 1432 core genes (see the sections "Datasets—inter-strain MSAs" and "Datasets—inter-species MSAs") widely present across *E. coli* strains. We also perform the same analysis on 1029 entire core gene sequences in order to increase site coverage. Results presented in the following sections are those obtained on Pfam domains, results on full-core genes are presented in Supplementary Figs. 1–5. The results for full sequences are mostly consistent but of lower quality than those obtained for Pfam domains, since the MSAs used for model training contain less and less diverse sequences.

DCA models provide a substitution score for each amino acid in each position, which depends on the sequence context of the protein domain in *E. coli*. On the contrary, the score of each amino acid in IND models is context-agnostic as it directly derives from its frequency of occurrence across distant homologs (see the sections "DCA and IND models" and "Individual mutation effect prediction by DCA and IND models"). To compare model predictions to reality, we gather a database of >60,000 *E. coli* strains where we record all polymorphisms occurring at frequencies >5%. We use a ST131 strain as a reference strain, this clonal complex is a public health concern because of its virulence and resistance to antibiotics[20] and has thousands of isolates sequenced in the database.

Amino acids observed in *E. coli* are well predicted by DCA, and to a lesser extent by IND. In all, 78% of amino acids observed in the reference strain rank first at their position with a DCA model while this figure drops to 45% with IND (Fig. 2a), in agreement with the previous study[16]. Approximately half of the time an

amino-acid site is polymorphic, the major allele is ranked first by DCA while minor alleles are more likely to rank second (Fig. 2b). Here again, DCA predictions overperform those of IND model (Fig. 2c). The DCA score distribution of *E. coli* polymorphisms centers on 0, meaning that DCA predicts them to be close to neutral (blue distribution, Fig. 2d). In comparison, DCA predicts that amino acids sampled from distant homologs and inserted in *E. coli* sequences will be deleterious (yellow distribution, Fig. 2d), a prediction IND cannot make. These results are consistent with the idea that mutations that fix in a population are close to neutral at the time they occur, but can be deleterious in another background. Figure 2d compares these scores with random mutations (gray histogram), predicting them to be even more deleterious since they include never observed mutations that are presumably highly counter-selected.

DCA and IND models predict mutation effects of amino-acid changes. However, the likelihood of observing an amino-acid change also depends on mutational biases. Among the 20 possible amino acids, we cannot obtain more than nine by mutating only one nucleotide of a given codon. On short evolutionary timescales, polymorphisms that require more than one single-nucleotide polymorphism (SNP) should rarely occur. If we set the probability of observing them to zero, the power to predict *E. coli* polymorphisms increases slightly but systematically for both models (by 5.3% for DCA and 11.0% for IND, Supplementary Fig. 6).

These results validate that even though DCA models are trained on distant homologs, they can capture the effect of natural selection at different timescales. Their ability to predict amino acids in the reference strain reflects the action of natural selection in fixing amino acids when *E. coli* diverged from other species. When it comes to predicting polymorphisms, it emphasizes the action of purifying selection on a shorter term. The better performance of DCA over IND highlights the major role played by epistasis in shaping mutation effect and the strong contingency of amino acids observed in *E. coli*. These results provide the support that DCA is an adequate tool to perform further studies in this work.

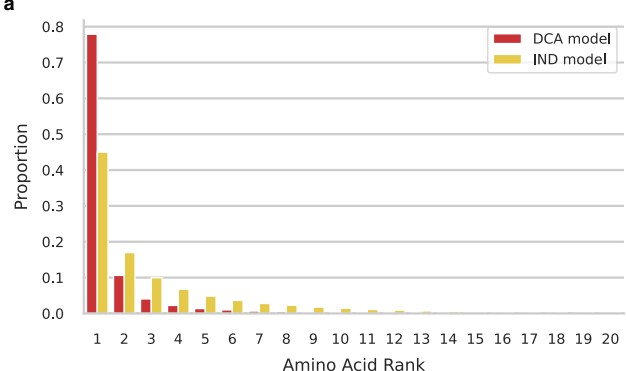

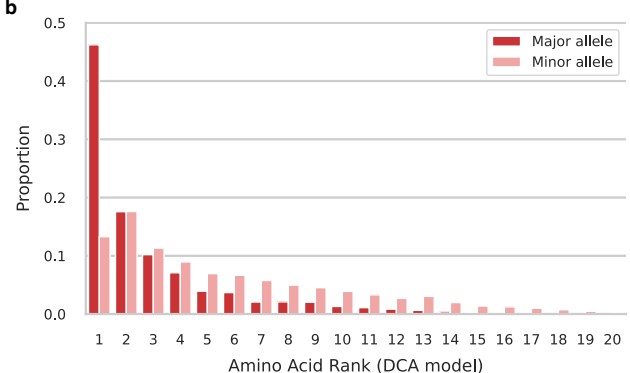

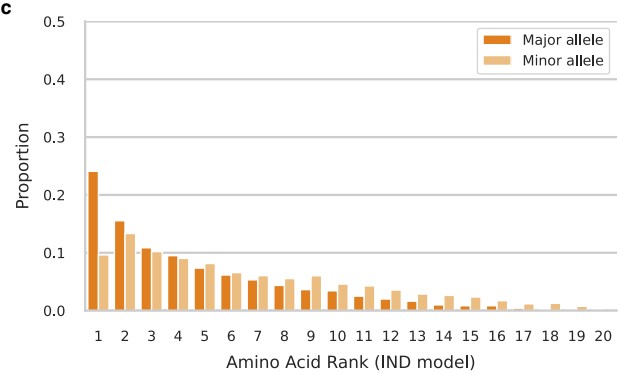

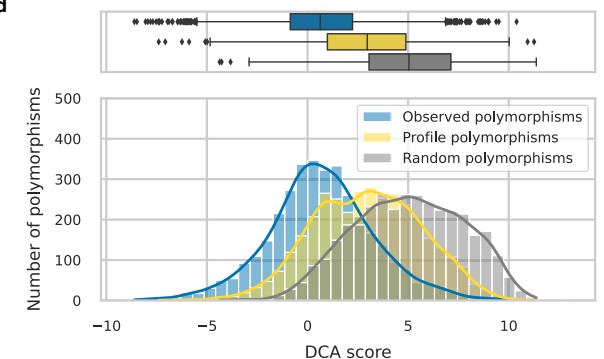

**Fig. 2 Predicted effects of observed amino acids using an IND model (neglecting epistasis) or a DCA model (incorporating pairwise epistasis).** **a** Rank of native amino acid in the reference strain as compared to all 20 possible amino acids. DCA model (red) outperforms IND (yellow) by predicting twice as many native amino acids to be the best possible. **b** DCA rank of major and minor allele for all sites that are polymorphic at a >5% threshold, among all 20 possible amino acids. Major alleles (alleles at frequencies >50%, in red) have better ranks than minor alleles (alleles at frequencies between 5 and 50%, in pink). The distribution of consensus alleles peaks at the first rank (46.2% of polymorphic sites have major allele ranking first and 17.6% have second-best rank) while the distribution of minor alleles peaks at the second rank (13.3% have the best rank against 17.6% that are second-best). **c** IND rank of major and minor allele for all sites that are polymorphic at a >5% threshold, among all 20 possible amino acids. As with DCA, major alleles (in orange) have better ranks than minor alleles (in yellow) and the distribution of consensus alleles peaks at the first rank. However, the distribution is spread towards greater ranks (only 24.1% of polymorphic sites have major allele ranking first and 15.5% have second-best rank, similarly minor alleles rank first in 9.6% and second-best in 13.3% of polymorphic sites) compared to DCA ranking. **d** Distribution of DCA scores of non-synonymous polymorphisms observed at frequencies >5% across the >60,000 strains (blue) compared to mutations sampled from an IND model (yellow) or to random mutations (gray). A large number of possible mutations are predicted to be highly deleterious (positive scores) compared to naturally occurring polymorphisms that tend to be neutral (blue distribution centered on zero). Polymorphisms predicted from IND are slightly deleterious once epistasis is taken into account (yellow distribution shifted towards positive values). Boxplot center lines represent medians, box limits are upper and lower quartiles, whiskers extend to show the rest of the distribution within an 1.5 × interquartile range, outliers are represented with points; sample size is 3477 mutations for each of the three groups.

When comparing protein sequences from distant species, we observe that some sites are conserved while others vary. However, if mutation effects depend on context, the level of variability observed at an amino-acid site across distant species may not reflect how polymorphic this site can be within any specific species.

We use Shannon entropy as an information-theoretic measure quantifying the diversity of amino acids observed at a given site (Fig. 3a). It measures the logarithm (in base 2) of the effective number of admissible amino acids at a position, if these were equiprobable. A site with an entropy of zero should only tolerate one amino acid: it is conserved. A value of one can for instance correspond to two amino acids at 50% frequency each. Entropy reaches its maximal value of $\log_2(20) = 4.32$, if all 20 possible amino acids are equally likely. Based on this concept, we can define a Context-Independent Entropy (CIE) from an IND model and an *E. coli* specific Context-Dependent Entropy (CDE) from a DCA model (see the section "Context-independent and context-dependent entropies").

We compute CIE at locus $i$ from the amino-acid frequencies $f_i(\beta)$ in the column $i$ of the MSA of distant homologs as:

$$\text{CIE}_i = -\Sigma_\beta f_i(\beta) \log_2 f_i(\beta) \tag{1}$$

where the sum is performed over all 20 amino acids $\beta$.

To compute CDE, we first need to determine the probability of observing a certain amino acid $\beta$ in position $i$, given that the other positions take amino acids $a_{\backslash i}^0 = (a_1, \ldots, a_{i-1}, a_{i+1}, \ldots, a_L)$ present in the *E. coli* reference sequence. Within our

**The sequence context constrains the predicted site variability in *E. coli*.** Focusing on individual amino acids, we have seen that native amino acids fixed in *E. coli* and polymorphisms observed in a wide collection of strains are strongly contingent on the genetic background. Going to an amino-acid site perspective, this raises the question of how much epistasis shapes site variability.

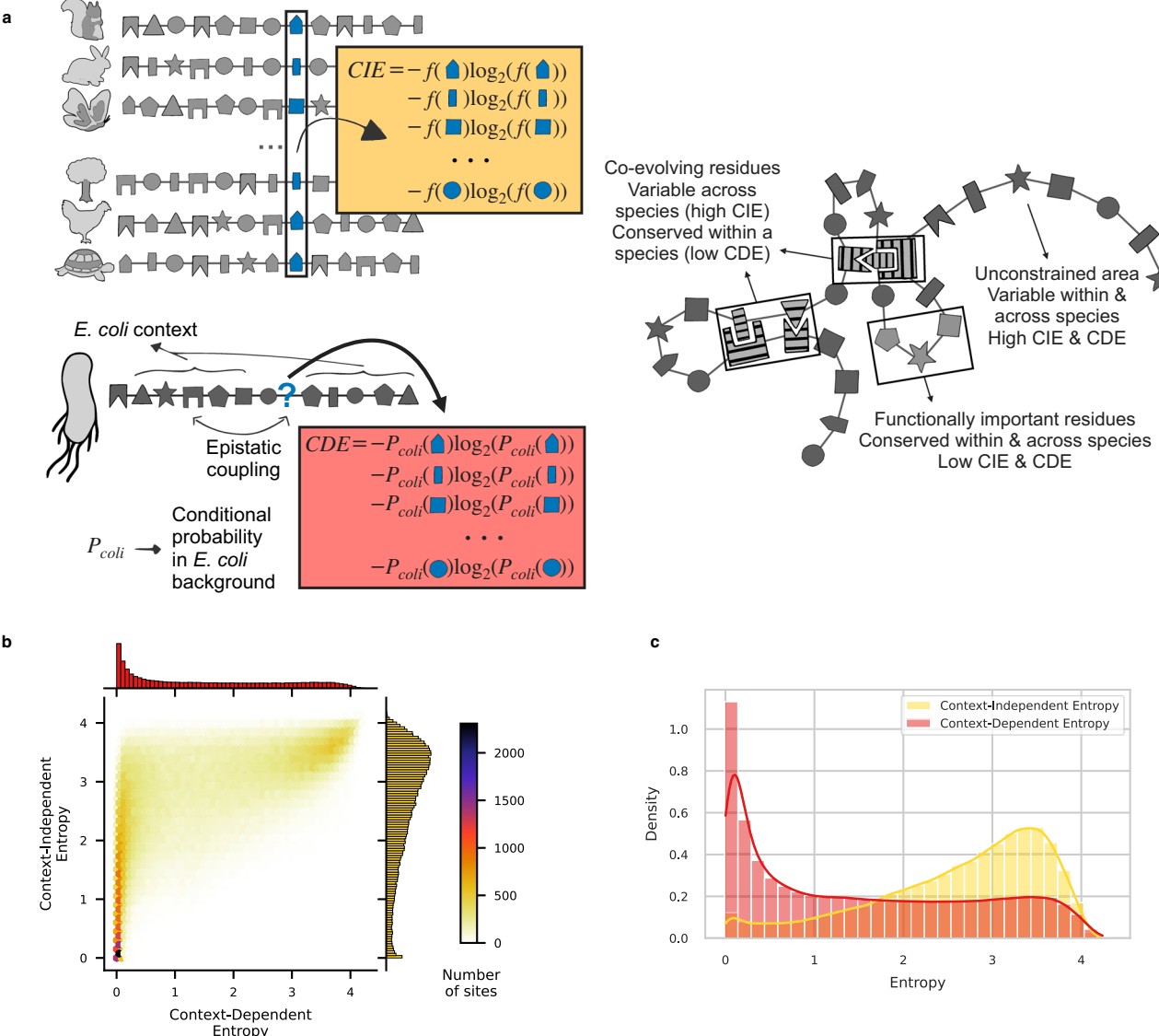

**Fig. 3 Predicting the variability of amino-acid sites. a** Entropy quantifies the level of variability of an amino-acid site from conserved (entropy ∼ 0) to highly variable (entropy ∼ 4). It can be computed from a non-epistatic model (Context-Independent Entropy (CIE), yellow) i.e., from the frequencies of amino acids observed across distant species, or from an epistatic model (Context-Dependent Entropy (CDE), red) i.e., from the conditional probabilities of observing each amino acid in *E. coli* background. Residues that have strong epistatic interactions with others will be lowly polymorphic once the genetic context is fixed (low CDE) but can vary between species (high CIE) by co-evolving with their partners (cf. hatched residues). **b** Bivariate histogram of CDE and CIE for all sites in the dataset. Two populations of sites are clearly recognizable, in particular separated by their CDE values. **c** Marginal distributions of CDE (red) and CIE (yellow) for all sites in the dataset. CDE divides amino-acid sites into two populations of similar sizes: conserved (CDE < 1) and variable (CDE ≥ 1). On the contrary, most of the amino-acid sites have a high CIE, i.e., IND predicts them to be highly variable.

DCA-based modeling framework, this quantity reads:

$$P_i(\beta|a^0_{\backslash i}) = \exp\left\{h_i(\beta) + \Sigma_{j \neq i} J_{ij}(\beta, a_j)\right\}/z_i, \qquad (2)$$

with the normalization $z_i$ chosen such that $P$ becomes a probability distribution over the values of $\beta$, i.e., over the 20 theoretically possible amino acids in position $i$ (gaps are not considered, since we study the effects of amino-acid substitutions and not deletions). CDE is now given by:

$$\text{CDE}_i(a^0_{\backslash i}) = -\Sigma_\beta P_i(\beta|a^0_{\backslash i}) \log_2 P_i(\beta|a^0_{\backslash i}), \qquad (3)$$

with $a^0_{\backslash i}$ being the sequence context of the *E. coli* reference strain.

CIE and CDE are both model-predicted quantities, that do not use any *E. coli* polymorphism data to predict variability within this species. CIE corresponds to the level of variability observed across distant species. CDE takes the amino-acid context and the local epistatic couplings of the reference strain into account to predict the level of variability within the *E. coli* sequence background. If epistasis were negligible, CIE and CDE values should be comparable.

Figure 3b shows a bivariate histogram of CIE and CDE over all sites in our dataset. Two distinct communities clearly emerge. On one side, a top-right peak of sites shows high CDE and CIE. These sites display very little context dependence (both entropies have comparable values). They reach entropy values near 4, i.e., close to the upper limit of $\log_2(20) = 4.32$. These sites are variable across distant species and predicted to be highly polymorphic in *E. coli*. On the other side, a left peak of sites has low CDE and low to high CIE. We predict them to be conserved in *E. coli* (CDE close to 0) but they can vary across distant species (CIE ranging

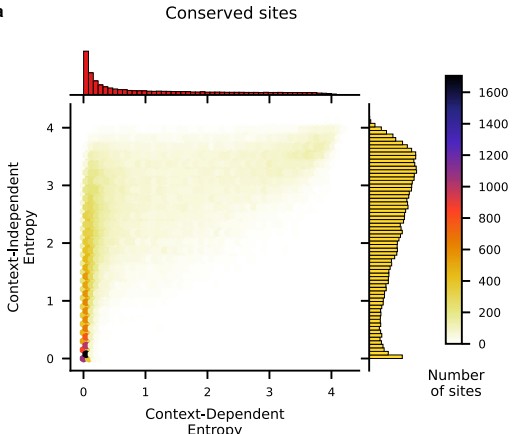

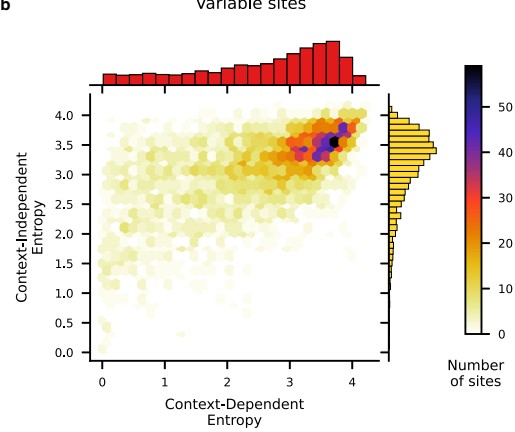

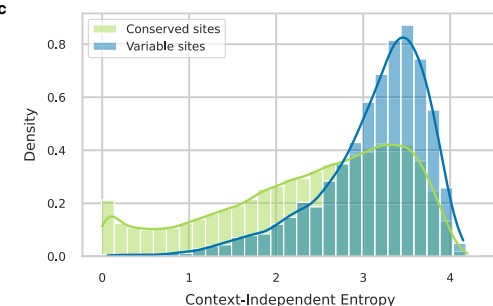

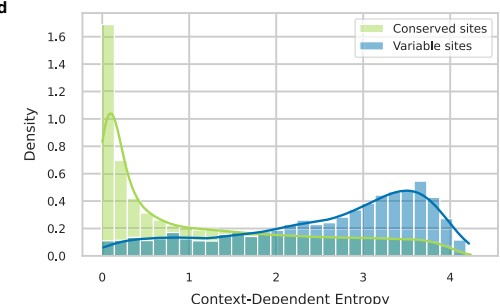

**Fig. 4 Predicting amino-acid sites that are conserved or polymorphic in *E. coli*. Comparison of the performance of IND and DCA models. a** Bivariate histogram of CDE and CIE for sites that are conserved across >60,000 strains of *E. coli*. Most of them cluster on the left peak of low CDE. **b** Bivariate histogram of CDE and CIE for sites that are polymorphic at a 5% threshold across >60,000 strains of *E. coli*. Most of them cluster on the right peak of high CDE. **c** Distribution of CIE for conserved (green) and polymorphic (blue) sites in *E. coli*. A non-epistatic model fails at distinguishing between both populations. Most of the sites are predicted to have a high entropy so to be highly variable, including those that display no mutation in >60,000 strains of *E. coli* (green distribution). **d** Distribution of CDE for conserved (green) and polymorphic (blue) sites in *E. coli*. A model that incorporates pairwise epistasis predicts a low entropy for conserved sites (the green distribution peaks near 0) and a high entropy for variable sites (the blue distribution peaks near 4).

**Context-dependent entropy accurately predicts polymorphic and constrained sites in *E. coli*.** We can now confront these model-based predictions to the observed variability in our dataset of >60,000 *E. coli* strains. To do so, we categorize *E. coli* sites into: conserved (no polymorphism observed in any of the strains) and variable (at least 5% of the strains harbor a mutation with respect to the consensus sequence).

Lowly polymorphic sites (<5%-frequency polymorphisms) can correspond to variable sites but also to conserved sites with deleterious mutations segregating at low frequencies (or sequencing errors for some of the lowest frequencies), so we choose to exclude them from the analysis.

Most of the conserved sites cluster on the left peak of low CDE (Fig. 4a) whereas variable sites tend to cluster on the top-right peak of high entropies (Fig. 4b). CDE appears more relevant than CIE to discriminate conserved from variable sites. Indeed, only 12.7% of conserved sites have CIE < 1 (Fig. 4c) while 56.4% have CDE < 1 (Fig. 4d). If we integrate mutational biases into our analysis, by restricting the computation of entropy to 1-SNP amino-acid mutations (see the sections "1-SNP mutations" and "Context-independent and context-dependent entropies"), we find that 70.2% of conserved sites have CDE < 1 whereas only 24.8% have CIE < 1 (Supplementary Fig. 7). Yet, there remain 29.8% of conserved sites that are predicted to be polymorphic (CDE ≥ 1). Looking at the synonymous diversity across the *E. coli* strains, we notice that many 1-SNP synonymous mutations are missing. This implies that only a limited amount of neutral diversity can segregate within a population, a limitation probably due to random drift. We thus use simulations based on the amount of observed synonymous diversity to estimate the proportion of sites we expect to see conserved while they could tolerate polymorphisms (high CDE) (see the section "Simulations of neutral diversity segregating on amino-acid sites"). These give results that are consistent with our observations (Supplementary Fig. 8): polymorphisms may arise on these sites but have not been observed in nature yet.

These results show that CDE accurately predicts the level of variability of an amino-acid site by integrating constraints linked to its function, common to all genetic backgrounds, and local epistatic couplings that are specific to a given genetic context. CIE misses most of the conserved sites, demonstrating how strongly the context reduces the variability, which is possible at an amino-acid site.

We now want to investigate how much the genetic context reduces the diversity of amino acids tolerated at a site. In other words, how contingent on the genetic background the effect of an amino-acid change is. Comparing CIE to CDE allows to quantify contingency, as they both measure site variability with CIE being

from 0 to more than 3). We expect these sites to display a low level of polymorphism across *E. coli* strains.

CIE and CDE distributions over all sites greatly differ (Fig. 3c). While only 8.3% of sites are conserved across distant species (CIE < 1, corresponding to an effective number of amino acids below 2), we predict 45% of sites to be conserved in *E. coli* (CDE < 1) largely due to local epistatic couplings.

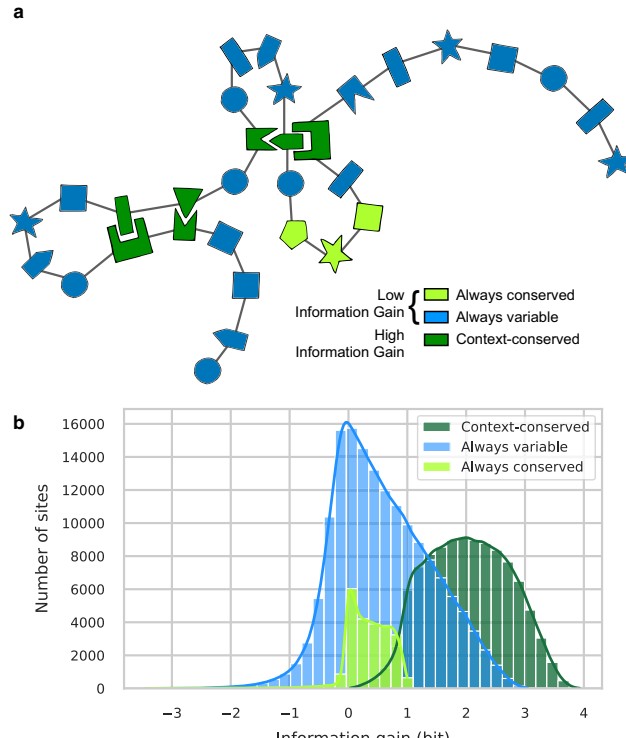

**Fig. 5 Quantifying the effect of the context in reducing amino-acid site variability. a** The genetic background is expected to differentially impact amino-acid sites. It has a low influence on sites that have the same level of variability in *E. coli* and across distant species (blue and light green). On the contrary, it strongly impacts sites that are variable across distant species but are conserved in *E. coli* due to local epistatic couplings (dark green). **b** Information gain quantifies the difference between an amino-acid site variability across distant species and its potential variability in *E. coli*. Sites that are variable across distant species (CIE ≥ 1) but conserved in *E. coli* (CDE < 1) are the ones with the highest information gains (dark green distribution). Note that the information gain is given in bits, 1 bit corresponds to an effective reduction of the available amino acids by a factor 2, 2 bits by a factor 4, and 3 bits by a factor 8.

context-agnostic and CDE being context-aware. We can split amino-acid sites into three categories (Fig. 5a). First, 8.3% of sites are conserved across all species as well as in *E. coli* (CIE < 1). They are likely to be functionally essential. Mutating away from the observed amino acid will always be deleterious, so the context has no real influence on their level of conservation. Second, 55.1% of sites are variable across all species as well as in *E. coli* (CIE ≥ 1, CDE ≥ 1). They are often constrained (CDE < $\log_2(20)$), but allow for a considerable amino-acid variability both in the family and in the specific *E. coli* context: at these positions, we may observe both fixed differences between species and polymorphisms within the *E. coli* population. Third, 36.6% of sites are conserved in *E. coli* context but variable across species (CIE ≥ 1, CDE < 1). Amino acids observed in distant species will not be tolerated in this specific context: evolution is contingent on the genetic background.

We define the information gain provided by the sequence context as the difference between CIE and CDE (see the section "Context-independent and context-dependent entropies"). If both are equal, no information is contained in the context. The lower CDE is compared to CIE, the greater the information gain and the level of contingency. We observe that the majority of sites have a positive gain in information when the sequence context is known (Fig. 5b). In 50.5% of sites, the effective number of acceptable

amino acids in the *E. coli* context is at least a factor two smaller than what a context-independent analysis of distant homologs would predict (information gain >1 bit). We conclude that roughly 30–50% of amino-acid sites show consistent signals of context dependence.

**Epistasis is a diffuse pattern involving a sum of many small couplings**. The higher accuracy of DCA over IND in predicting site variability and amino acids observed in *E. coli* proves that epistasis strongly shapes the effect of mutations. Following this observation, we want to use DCA as a tool to study epistasis in natural isolates. First, we look at epistasis between polymorphisms arising jointly in *E. coli*. To do so, we gather all gene sequences with exactly two amino-acid substitutions (other than gaps, i.e., deletions or insertions) compared to the reference strain. For each pair of mutations, we compare the DCA-predicted effect of the double mutation to the sum of the effects of each single mutation introduced individually in the reference sequence ("Epistatic cost"). We observe no clear difference between these two quantities (Fig. 6a), indicating an absence of strongly coupled polymorphisms. Two main factors may explain the absence of strong epistatic couplings between polymorphisms in *E. coli*. First, polymorphisms arise on highly variable sites: these sites are poorly constrained by epistasis (high CDE). Second, previous works claim that epistasis is often weak compared to the typical effect size of mutations[21]. This second point does not contradict the strong context dependence of mutations. It suggests that context might be a collective effect arising from the accumulation of many small epistatic couplings. Importantly, these couplings may involve sites that are conserved in *E. coli* but vary across distant species. We use inverse participation ratio (IPR)[22] to estimate the proportion of sites effectively coupled to a locus in amino-acid sequences modeled with DCA (Fig. 6b and section "Effective proportion of residues coupled to an amino-acid site"). IPR allows one to determine the effective number of non-zero components of a distribution. This effective number is minimal in case of a single one non-zero component, and maximal for a uniform distribution with identical entries. We find that each amino-acid site is coupled to about one-fourth of the rest of the protein. Taken altogether, these results lead us to consider that context dependence of mutations does not rely on a few strong epistatic couplings but on an aggregation of many small couplings accumulated with divergence.

**Gradual construction of the context with divergence**. So far, we have gathered evidence that many small couplings accumulate to build a genetic context. This translates into an absence of a strong epistatic signature of polymorphisms co-occurring in *E. coli*. However, we expect epistasis patterns to emerge gradually when the number of substitutions increases. To study how the genetic background is building up with divergence, we gather 853 Pfam domains spanning 516 core genes shared by diverged species from *E. coli* to *Yersinia pestis* (Fig. 7a and section "Datasets— closely diverged species MSAs").

We start by comparing pairs of homologous sequences. For each pair, we compute the DCA epistatic cost as being the difference between the DCA score of the fixed differences altogether and the sum of their DCA effects when inserted individually in one of the two genetic backgrounds (see the section "Epistatic cost"). It is worth noting that a negative DCA epistatic cost corresponds to positive epistasis: fixed differences are more beneficial, i.e., have a lower DCA score, taken altogether than expected by the sum of their individual effects. As gaps can artificially create a pattern of positive epistasis, we only keep pairs of sequences that have no more than one gap difference. We observe a clear pattern of positive epistasis that

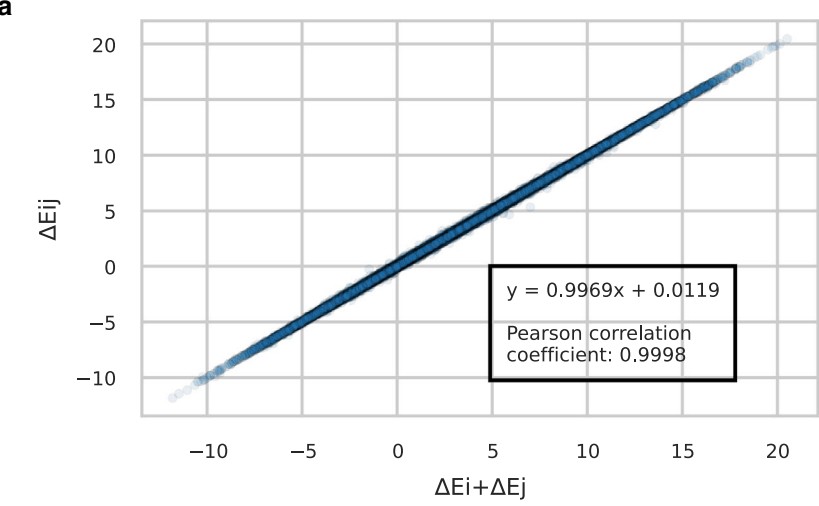

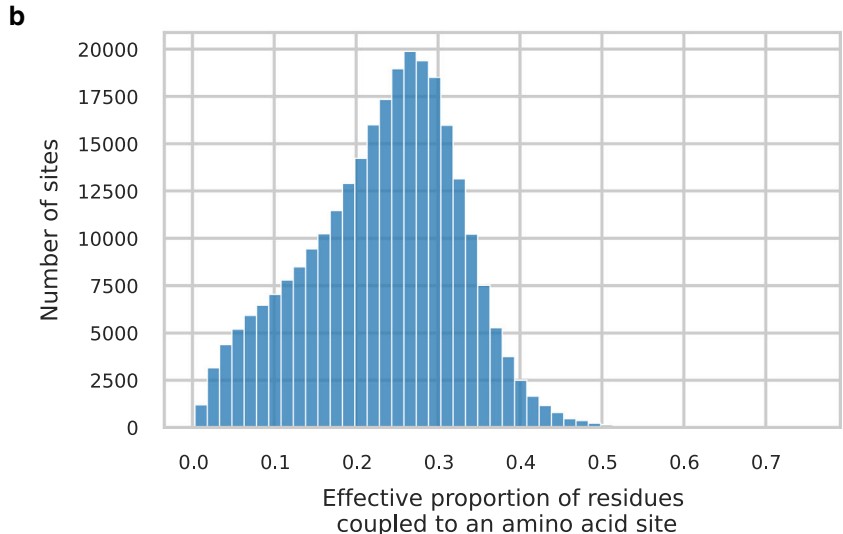

**Fig. 6 Epistasis observed in *E. coli*. a** Mutational effect $\Delta E_{ij}$ of observed double mutations with respect to the reference, plotted against the sum $\Delta E_i + \Delta E_j$ of the individual mutation scores. The absence of clear deviations from the diagonal reveals the lack of strong epistatic couplings between pairs of mutations in our strain dataset. **b** Histogram of the effective proportion of sites coupled with a given amino acid. It is computed from the inverse participation ratio: $1/(\text{IPR} \times \text{proteinlength})$. The median of the distribution is 24%, meaning that amino-acid sites are generally coupled to about one-fourth of the other residues in the protein according to DCA modeling of epistasis.

increases with divergence (Fig. 7b). This is consistent with a model where fixed differences are contingent on previous mutations and entrenched by subsequent ones. Individual couplings are biased towards positive epistasis (pronounced left tail of negative DCA couplings between pairs of fixed differences in Fig. 7c). However, their values rarely fall below -1 (note the log scale of the vertical axis), a rather low effect size compared to the most extreme epistatic costs that can be measured between entire sequences in Fig. 7b. This is consistent with epistatic patterns emerging gradually by addition of small couplings accumulated with divergence. The more diverged the sequences, the stronger the epistatic signal because each additional fixed difference modifies many couplings. These sequences have evolved naturally since their corresponding species diverged: the over-representation of positive epistatic couplings that we detect is consistent with evolution under long-term purifying selection[4].

***rplK*: a gene displaying a strong epistatic signal**. *rplK* codes for the L11 protein of 50S subunit of the ribosome. It exhibits a

strong signal of positive epistasis among the 14 non-synonymous mutations fixed between *E. coli* and *Y. pestis*. This relatively small number of fixed differences offers a good opportunity to investigate how epistasis emerges at an individual protein level.

The range of epistatic couplings between fixed differences (Fig. 8a) is consistent with Fig. 7c: no very strong couplings but a clear tendency towards negative DCA values (i.e., positive epistasis). The strongest epistatic couplings correspond to pairs of residues that are in close vicinity in the 3D folding of the protein (distances <10 Å in Fig. 8b). We also observe a clear over-representation of couplings near −0.2—as compared to the number of couplings near 0.2—the majority of which correspond to more distant pairs of sites. Even if these residues are not necessarily in contact with one another, almost all of them cluster in the protein structure (red spheres in Fig. 8c). This suggests that epistasis does not solely arise from direct contacts between few neighboring residues but also from more distant interactions between amino acids that contribute to the stability of the protein structure. We previously found that DCA predicts about one-fourth of amino-acid sites to be effectively coupled to a given

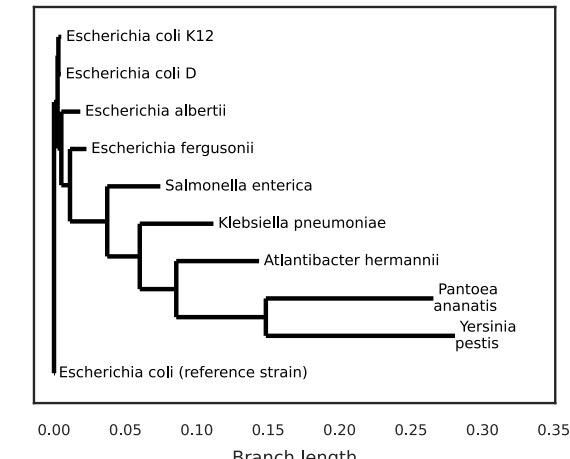

**a**

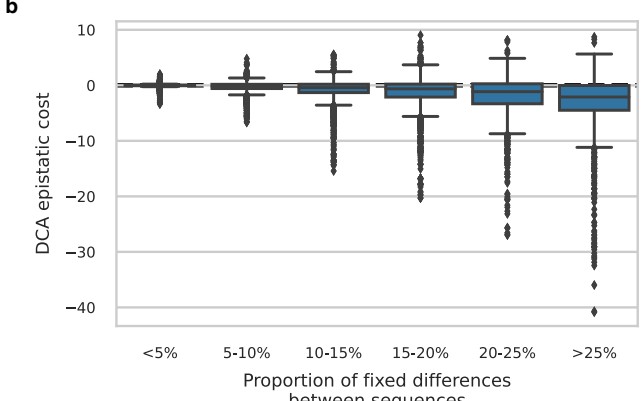

**b**

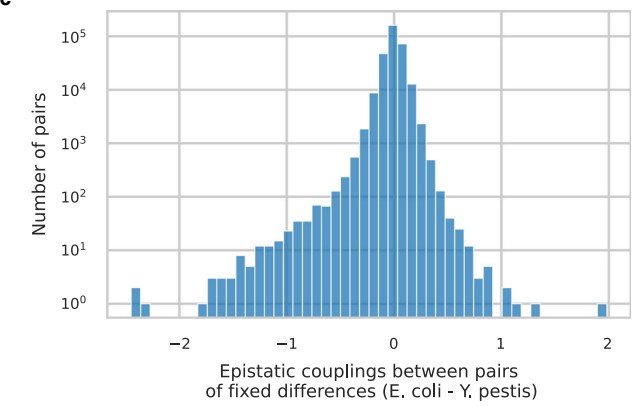

**c**

**Fig. 7 Epistasis between fixed differences in a panel of diverged species. a** Phylogenetic tree of studied strains. Tree built from an amino-acid sequence alignment of 878 core genes. **b** DCA epistatic cost decreases with divergence. It is defined as the difference between the total change in statistical energy between pairs of sequences and the sum of single-mutant effects. Negative values correspond to positive epistasis: mutations are more beneficial (lower DCA score) taken altogether than the sum of their individual effects. Boxplot center lines represent medians, box limits are upper and lower quartiles, whiskers extend to show the rest of the distribution within an 1.5 × interquartile range, outliers are represented with points. Sample sizes are $n = 22{,}352$ for <5%, $n = 15{,}870$ for 5−10%, $n = 10{,}810$ for 10−15%, $n = 6776$ for 15−20%, $n = 3564$ for 20−25%, $n = 3432$ for >25%. **c** Distribution of epistatic couplings between pairs of fixed differences between *E. coli* and *Y. pestis*. The distribution is shifted towards negative values corresponding to positive epistatic couplings between fixed differences: they are better together than the sum of their individual effects. The relative small values of these couplings as compared to overall epistatic scores measured between entire sequences (**b**) indicate that epistatic patterns build up gradually by an accumulation of many small couplings.

residue. This figure clearly exceeds the number of residues that are in physical contact with an amino-acid site but could be explained by the hypothesis that sites belonging to the same protein domain are epistatically coupled with one another even if not in direct contact. These domains of correlated residues that co-evolve over long evolutionary times are reminiscent of protein sectors[23]. They are also consistent with recent experimental work showing that DCA couplings can capture global phenomena such as allosteric communication between DNA-binding and ligand-binding modules in a protein[24].

## Discussion

The adaptationist and neutralist interpretations of biological diversity have long neglected epistasis. The complexity of modeling epistasis certainly contributes to explaining why

independent-site models remain common in molecular evolution. Breen et al. first raised the possibility of epistasis being "the primary factor" in protein evolution[5]. Even if their methodology based on dN/dS computations underwent criticism[25], it clearly called for a deeper and more systematic study of epistasis across the genome. Experimental studies of mutations in different genetic backgrounds have confirmed an important role of epistasis in long-term evolution[7,8]. However, they remain constrained to the analysis of single proteins. As abundant genetic data for both *E. coli* strains and diverged species have become available, data-driven approaches offer new opportunities. Through the concept of DCA-informed amino-acid landscapes, this allows for a large-scale data-driven study of epistasis on both short- and long-term evolution. The systematic analysis of wide genome portions has the potential to unveil much more widespread mechanisms than the potentially idiosyncratic studies led on specific proteins.

We find that DCA overperforms IND in predicting native amino acids as well as observed mutations and amino-acid site variability within *E. coli* strains. Intriguingly, DCA also ranks major and minor alleles better than the IND model, suggesting that epistasis can constrain variable sites. Native amino acids arise from long-term evolution whereas observed polymorphisms and site variability within *E. coli* strains reflect short-term evolution. Thus, amino-acid landscapes appear relevant to study both short- and long-term evolution even though they are inferred from highly diverged species and can only capture evolutionary forces that are conserved for the entire family. Interestingly, it suggests that local adaptation of some specific strain to some specific ecological niche might add on top of these general constraints but does not dominate evolution. Our data analysis also emphasizes the importance of mutational biases on short evolutionary timescales. Neutral polymorphisms that require more than one SNP are virtually absent.

The better performance of DCA as compared to IND demonstrates the importance of taking epistasis into account to understand the effect of amino-acid changes. Recent achievements in synthetic biology prove that DCA captures enough protein constraints to predict functional variants having less than 65% identity with amino-acid sequences used to train the DCA model[13]. They also experimentally demonstrate that an IND model fails at generating functional variants. This leads us to question the widespread use of software based on independent-

**a**

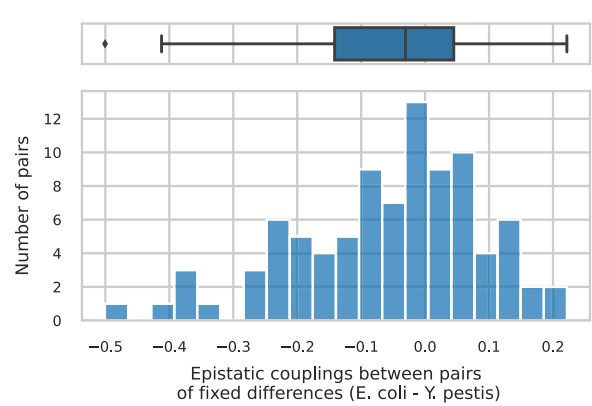

**b**

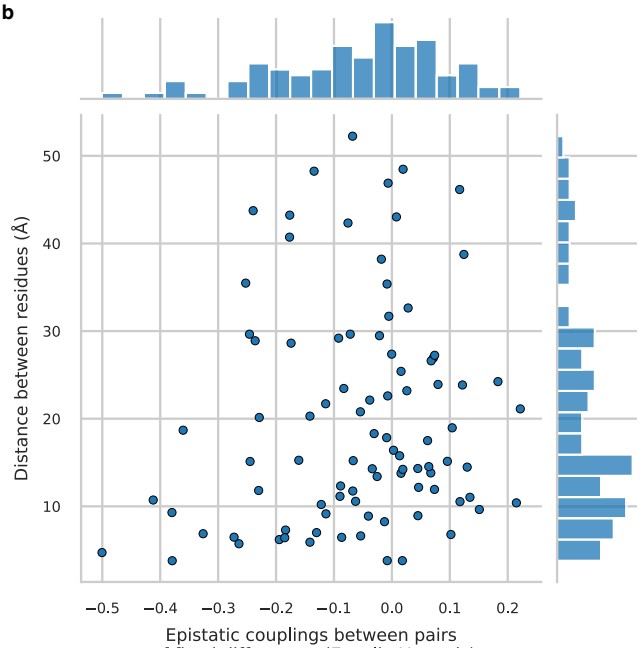

**c**

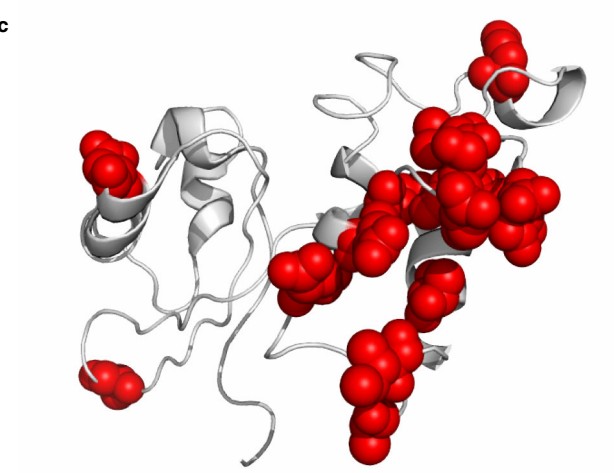

**Fig. 8 Epistatic couplings between amino-acid differences that have fixed between *E. coli* and *Y. pestis* in *rplK* gene. a** Distribution of epistatic couplings between pairs of fixed differences. The left tail of negative DCA scores signals an over-representation of positive epistatic couplings. Boxplot center line represents the median, box limits are upper and lower quartiles, whiskers extend to show the rest of the distribution within an 1.5 × interquartile range, outliers are represented with points, sample size is $n = 91$ couplings. **b** Joint distribution of epistatic couplings values between pairs of residues harboring a fixed difference and their physical distance in the 3D structure of the protein. The strongest couplings correspond to residues that are in contact (<10 Å). However, most of the couplings involve residues that are more distant than 10 Å. **c** Representation of the 3D structure of the protein encoded by *rplK*: residues that differ between *E. coli* and *Y. pestis* are highlighted with red spheres. Most of the fixed differences cluster together in the same domain, explaining why we observe a strong epistatic signal even though most of the pairs of fixed differences are not in physical contact.

site models such as SIFT[26] or Polyphen[27] to predict mutation effects. Here, we use DCA to characterize *E. coli* evolutive history. However, it paves the way to a far broader range of applications such as predicting adaptation or understanding molecular mechanisms underlying genetic diseases. In the latter case, DCA may prove useful at investigating cases of Dobzhansky–Muller incompatibilities[28] where amino-acid changes that have been fixed in distant species would be pathogenic to humans. For more applied purposes, DCA could be used to single out causative mutations associated to diseases in human genetics.

In agreement with ref. [5], we find that context dependence dramatically reduces the variability observed at a given amino-acid site. Epistasis, therefore, plays an important role in evolution. However, we show that epistatic couplings between pairs of sites remain small compared to the typical effect of a mutation. Our data suggest that the strong context dependence of mutation effect comes from an accumulation of many small couplings. Consequently, most of the polymorphisms that arise within a species should have the same effect in all strains: the amino-acid landscape near a reference strain is locally smooth. In contrast, the global landscape is rougher, with about one-third of amino-acid sites where the effect of mutations drastically varies between distant species. Analyzing a panel of closely diverged species through DCA modeling, we are able to show how these epistatic patterns gradually emerge with divergence.

Deep mutational scans have shown that positive epistasis between pairs of amino acids is less common than negative epistasis[3]. However, we show that positive epistatic couplings between residues dominate long-term evolution. Simulating the evolution of ArgT protein, Shah et al. have already noticed that, under purifying selection, mutations that fix are enriched in positive epistatic couplings with the rest of the background[4]. This is because purifying selection favors both mutations that are beneficial in all backgrounds and mutations that are beneficial in a given background due to epistatic couplings with the rest of the sequence. Here, we observe the same phenomenon with real data and across hundreds of genes. Quantifying these effects experimentally would require performing deep mutational scans on several homologs at different distances with extremely accurate fitness estimates to detect small effects.

According to our findings, polymorphisms currently occurring in *E. coli* are close to neutral. On the contrary, fixed differences with *Y. pestis* tend to be deleterious in *E. coli* background. These observations perfectly fit a scenario of contingency and entrenchment: mutations are neutral at the time when they appear while being contingent on previous mutations and entrenched by subsequent mutations[4]. However, our approach to analyzing context dependence is necessarily limited by the accuracy of DCA at modeling epistatic interactions. We have gathered evidence that DCA correctly captures the local neighborhood near *E. coli* sequences. These results combined with other assessments of DCA predictive power[12,13] lead us to believe

that it should be informative on how context dependence evolves with divergence. We cannot, though, reject the hypothesis that some of our observations are not a true biological signal but more artifacts of DCA modelings. In particular, DCA may capture some phylogenetic correlations as well as true epistatic couplings. In fact, accidental cooccurrences of mutations along the branches of a phylogeny have been previously shown to generate non-trivial correlations between residue positions[29], which in turn lead to non-zero, but spurious couplings in DCA models[30], overlaying the true epistatic couplings. An analysis of the impact of the phylogeny (see Supplementary Notes) in our dataset shows that, as expected, phylogeny-induced spurious couplings result in lower site entropies than in independent models (Supplementary Fig. 10a). However, full DCA models capture more couplings and have therefore even lower site entropies. In addition, the artifactual couplings created by phylogeny worsen our ability to predict observed polymorphic and conserved sites in *E. coli* (Supplementary Fig. 10b), proving that phylogeny cannot explain the patterns we observe across strains.

DCA model performance relies on the quality of the interspecies MSAs on which models are learned. Pfam-domain MSAs are deeper and more diverse than full-protein MSAs because many different proteins across a wide range of organisms can share the same Pfam domain. As a consequence, DCA models trained on Pfam-domain MSAs overperform those trained on full-protein MSAs in predicting native amino acids and mutation effects (Supplementary Figs. 1–5). However, full-protein MSAs cover a larger fraction of the genome, and DCA models trained on them perform well at predicting site variability. The choice of the MSA reveals a trade-off between the DCA model accuracy and the fraction of the genome that can be covered. Depending on the intended applications, one might be favored over the other.

Since landscape models are inferred one by one for each protein, we can only capture intraprotein epistasis, but not any epistatic interaction between proteins. This is not an intrinsic limitation of the DCA approach, epistatic landscapes connecting two or more proteins may be inferred from joint MSAs[31]. However, the size of the model grows quadratically with the number of amino-acid sites, making the inference of a full joint core genome landscape impractical in terms of computational time. Even by restricting to intraprotein epistasis, we obtain amino-acid landscapes that are relevant to study evolution on short and long timescales. The substantial context dependence of mutation effects that we uncover may be enhanced by accounting for inter-protein epistasis.

## Methods

**Datasets—interstrain MSAs.** In all, 61,157 *E. coli* genomes are downloaded from Enterobase[32]. In total, 298,781,787 coding sequences are detected by Prokka 1.13.3[33]. In all analyses, the reference strain is the GA4805AA genome (available on NCBI[34] under BioProject accession id PRJNA218163). For each gene in the reference strain, homologous sequences in the other genomes are retrieved using phmmer from HMMER 3.3.1[35] (parameters: --popen 0.0001 --pextend 0.01) followed by a curation step where only sequences with less than 10 gaps after being aligned on the reference and more than 90% identity with the reference are kept. All genes with at least 60,000 homologous sequences are kept, these are referred to as core genes. Amino-acid sequences are aligned using mafft v7.471[36] and DNA sequences are reverse-aligned from amino-acid sequence alignments to preserve codon alignments. Two types of multiple-sequence alignments (MSAs) are generated: one with the full-length core gene sequences (full gene MSAs, produced for genes that are present in at least 61,000 genomes) and one per Pfam domain[19] present in a core gene (Pfam-domain MSAs).

**Datasets—closely diverged species MSAs.** The coding sequences of nine genomes of species closely related to *E. coli* are downloaded from Mage[37]: *Escherichia coli*K12 - chromosome ECK.1, *Escherichia coli*UMN026 - chromosome ESCUM.2, *Escherichia albertii*TW07627 - chromosome ESCAL.1, *Escherichia fergusonii*ATCC 35469T - chromosome EFER.2, *Salmonella enterica*subsp. arizonae serovar 62:z4,

z23:-- RSK2980 - chromosome NC\_010067.1, *Klebsiella pneumoniae*1162281 - WGS AFQL.1, *Atlantibacter hermannii*4928STDY7071316 - WGS CABGLB01.1, *Pantoea ananatis*AJ13355 - chromosome NC\_017531.1, *Yersinia pestis*Angola - chromosome NC\_010159.1. Homologous sequences are retrieved using vsearch v2.15.1[38] usearch_global command against the reference genome (parameters: --strand plus --id 0.5 --query_cov 0.8 --target_cov 0.8 --maxaccepts 100). Only core genes (genes with a homolog in all 9 genomes) are kept. Amino-acid sequences are aligned by mafft v7.471[36]. Both full-gene MSAs and Pfam-domain MSAs are generated. Full genes MSAs are also concatenated to produce a unique MSA used to generate a phylogeny with FastTree 2.1.3[39].

**Datasets—interspecies MSAs.** For each full-gene interstrain MSA and full-gene closely diverged species MSA, the corresponding full-gene interspecies MSA is produced by querying the corresponding reference amino-acid sequence against UniRef30 2020-03[40] using HHblits 3.3[41] followed by a curation step where sequences with more than 10% gap are removed from the MSA.

For each Pfam-domain interstrain MSA and Pfam domain closely diverged species MSA, the corresponding Pfam-domain interspecies MSA is generated by downloading the full Pfam alignment from the Pfam 34.0 (March 2021) database[19] and aligning the reference sequence to the Pfam HMM using hmmalign from HMMER 3.3.1[35]. All sites corresponding to inserts in the reference sequence are removed from the reference sequence, sites that are gapped in the reference sequence after aligning it to the Pfam HMM are removed from the Pfam MSA.

**DCA and IND models.** Direct-Coupling Analysis in the pseudolikelihood maximization framework (plmDCA)[42] is used to train DCA models, using standard settings: $\theta = 0.2$ (for reweighting) and $\lambda_J = 0.01$, $\lambda_h = 0.01$ (for L2 regularization).

For each interstrain MSA, the corresponding interspecies MSA is filtered to remove all sequences with >90% identity with the reference sequence. A DCA model is then trained if the filtered interspecies MSA contains more than 200 sequences. While this may appear a low threshold, know that most Pfam MSAs are much larger with an average size of 50,988 sequences, and more than 95% of Pfam MSAs that have at least 913 sequences (Supplementary Fig. 9).

For each closely diverged species MSA, a tree is built with FastTree[39] from the corresponding interspecies MSA concatenated to the closely diverged species MSA. The most recent common ancestor to the closely diverged species is inferred from this phylogeny. Any sequence of the interspecies MSA that descends from this most recent common ancestor is removed from the interspecies MSA. This is done in order to limit the risk of phylogenetic couplings to interfere with true epistatic interactions when training DCA models. A DCA model is then trained if the filtered interspecies MSA contains more than 200 sequences.

Each time a DCA model is trained, a corresponding IND model is produced from the frequencies of all possible amino acids or gaps at each position in the filtered interspecies MSA used to train the DCA model. Frequencies are computed after a reweighting step ($\theta = 0.2$) to give similar weights to training sequences than in the DCA model. The reweighting step is performed using DCAUtils.

**Individual mutation effect prediction by DCA and IND models.** When no particular software is mentioned, analyses are performed using Python3 v3.8[43] and Biopython v1.77[44]. Amino-acid sites that are gapped in more than 20% of the sequences of the interspecies or intra-species MSAs are never considered.

A DCA model trained on an interspecies MSA of length $L$ is composed of two matrices: $h$ and $J$. They can be used to assign a statistical energy $E(a_1, …, a_L)$ to any amino-acid sequence $(a_1, …, a_L)$:

$$E(a_1, … , a_L) = -\Sigma_{i<j} J_{ij}(a_i, a_j) - \Sigma_i h_i(a_i), \tag{4}$$

The $h_i(a_i)$ are site-dependent biases taking into account the importance of single amino acids in individual sequence positions; the $J_{ij}(a_i, a_j)$ are epistatic couplings connecting the amino acids in pairs of positions. The function $E$ is inferred to maximize the pseudolikelihood of the sequences in the interspecies MSA.

Two amino-acid sequences can be compared to one another by simply making the difference between their statistical energy values. In particular, the DCA score of mutating amino acid $\alpha$ into amino acid $\beta$ at position $i$ in the amino-acid background $(a_1, …, a_{i-1}, a_{i+1}, …, a_L)$ is given by:

$$\Delta E_i = E(a_1, … , a_{i-1}, \beta, a_{i+1}, … , a_L) - E(a_1, … , a_{i-1}, \alpha, a_{i+1}, … , a_L)$$
$$= h_i(\alpha) - h_i(\beta) + \Sigma_{j\neq i} J_{ij}(\alpha, a_j) - \Sigma_{j\neq i} J_{ij}(\beta, a_j), \tag{5}$$

The DCA score of the mutation $\alpha \to \beta$ at locus $i$ in the amino-acid background $(a_1, …, a_{i-1}, a_{i+1}, …, a_L)$ can be turned into a conditional probability of observing the amino acid $\beta$ at locus $i$, given that the other positions take amino acids $a^0_{\backslash i} = (a_1, … , a_{i-1}, a_{i+1}, … , a_L)$. Within our DCA-based modeling framework, this quantity is given by Eq. (2):

$$P_i(\beta|a^0_{\backslash i}) = \exp \left\{ h_i(\beta) + \Sigma_{j\neq i} J_{ij}(\beta, a_j) \right\} / z_i$$

with the normalization $z_i$ chosen such that $P$ becomes a probability distribution over the values of $\beta$, i.e., over the 20 theoretically possible amino acids in position $i$ (gaps are not considered, since we study the effects of amino-acid substitutions and no deletions).

The probability of observing amino acid $\beta$ at locus $i$ in IND is given by the frequency of amino acid $\beta$ at locus $i$ in the interspecies MSA (after sequence reweighting, see the section "DCA and IND models"): $f_i(\beta)$.

**Context-independent and context-dependent entropies**. The Context-Independent Entropy (CIE) is the standard column entropy of the interspecies MSAs. It is calculated from the position-specific amino-acid frequencies $f_i(\beta)$, measuring the fraction of sequences in the interspecies MSA having amino acid $\beta$ at locus $i$, using Eq. (1):

$$\mathrm{CIE}_i = -\Sigma_\beta f_i(\beta) \log_2 f_i(\beta).$$

The Context-Dependent Entropy (CDE) is computed from the conditional probabilities of observing the amino acid $\beta$ at locus $i$ in the amino-acid context of the reference strain $P_i(\beta|a^0_{\backslash i})$ with the formula of Eq. (3):

$$\mathrm{CDE}_i(a^0_{\backslash i}) = -\Sigma_\beta P_i(\beta|a^0_{\backslash i}) \log_2 P_i(\beta|a^0_{\backslash i})$$

The difference between CIE and CDE gives the information gain (IG) provided by the context:

$$\mathrm{IG}_i(a^0_{\backslash i}) = \mathrm{CIE}_i - \mathrm{CDE}_i(a^0_{\backslash i}), \qquad (6)$$

**1-SNP mutations**. All codons in the reference genome are analyzed in order to record all possible synonymous mutations and non-synonymous mutations that can be obtained by mutating them exactly once. These mutations are referred to as 1-SNP mutations. For non-synonymous mutations, the corresponding amino acids encoded by the mutated codons are also recorded.

The probability of observing an amino acid $\beta$ can be computed from an IND model restricted to 1-SNP mutations, by setting to 0 all entries of the $f_i(\beta)$ vector that do not correspond to 1-SNP mutations and re-normalizing $f_i(\beta)$. These new probabilities can be used to compute a CIE that is restricted to 1-SNP mutations.

The probability of observing an amino acid $\beta$ can be computed from a DCA model restricted to 1-SNP mutations, by setting to 0 all entries of the $P_i(\beta|a^0_{\backslash i})$ vector that do not correspond to 1-SNP mutations and re-normalizing $P_i(\beta|a^0_{\backslash i})$. These new probabilities can be used to compute a CDE that is restricted to 1-SNP mutations.

**Simulations of neutral diversity segregating on amino-acid sites**. Simulations are used to estimate the amount of neutral diversity segregating in *E. coli*. They are performed in two steps:

1. A calibration step where synonymous mutations are drawn from a Poisson distribution of parameter $\lambda$. The $\lambda$ value that best fit the observed amount of synonymous mutations is selected.
2. A simulation of genome evolution where both synonymous and non-synonymous mutations are sampled using the selected $\lambda$ value and can maintain in the population depending on their fitness cost (synonymous mutations are supposed to be neutral and DCA score is used as a proxy for fitness cost of non-synonymous mutations).

All simulations are based on Jukes–Cantor model (JC69). Only sites where the reference codon is the major allele are considered.

The calibration step is led on codons for which exactly three synonymous 1-SNP mutations are possible. A random number $N$ is sampled from a Poisson distribution of parameter $\lambda$: it corresponds to the total number of synonymous mutations occurring at this site. $N$ codons are then sampled with replacement from the three synonymous mutations possible at this site (with equiprobability). Each of these codons is kept with an acceptable probability of 50%. The number of different codons that are accepted at each site is recorded. Its minimal value is one (the reference codon alone) and the maximal value it can take is four (the reference codon and all three others synonymous mutations). Twenty simulations for each $\lambda$ ranging from two to five with a 0.1 step size are run to select the value of $\lambda$ for which the average number of synonymous mutations per site is the closest to what is observed in the >60,0000-strain dataset.

The simulation of genome evolution is then performed for all the sites of the dataset, excepting those for which the reference codon is not the major allele. For each site, a total number of mutations, $N$, is sampled from a Poisson distribution of parameter $\lambda$ (using the $\lambda$ estimated during the calibration step). $N$ codons are sampled with replacement from the nine possible codons (with equiprobability). Each of these codons is kept with an acceptance probability p = P(observing derived amino acid at locus $i|a^0_{\backslash i}$)/(P(observing derived amino acid at locus $i|a^0_{\backslash i}$) + P(observing reference amino acid at locus $i|a^0_{\backslash i}$)), where P(observing a given amino acid at locus $i|a^0_{\backslash i}$) is the conditional probability of observing this amino acid at locus $i$ given the amino-acid context of the reference strain, computed with DCA.

**Epistatic cost**. Epistasis is defined as the deviation from additivity of mutational effects. Having two mutations in sites $i$ and $j$ of a protein, the total mutational effect $\Delta E_{ij}$, defined as the difference in statistical energy between the double mutant and the reference sequences, can be compared to the sum $\Delta E_i + \Delta E_j$ of the effects of the two single-site mutations, individually inserted into the reference sequence. The

epistatic cost for substituting the reference residues $\alpha_i$, $\alpha_j$ with $\beta_i$, $\beta_j$ is the difference:

$$\Delta\Delta E_{ij} = \Delta E_{ij} - \Delta E_i - \Delta E_j = J_{ij}(\alpha_i, \beta_j) + J_{ij}(\beta_i, \alpha_j) - J_{ij}(\beta_i, \beta_j) - J_{ij}(\alpha_i, \alpha_j), \quad (7)$$

Similarly, the epistatic cost of an arbitrary number of mutations is the difference between the total mutational effect $\Delta E_{ij\ldots n}$ of the mutations altogether (i.e., the difference in statistical energy between the mutant and the reference sequences) and the sum $\Delta E_i + \Delta E_j + \ldots + \Delta E_n$ of the effects of the all single-site mutations, individually inserted into the reference sequence:

$$\Delta\Delta E_{ij\ldots n} = \Delta E_{ij\ldots n} - \Delta E_i - \Delta E_j - \ldots - \Delta E_n, \qquad (8)$$

For each interstrain MSA, sequences with exactly two mutations compared to the reference sequence and no gap are gathered. The total mutational effect $\Delta E_{ij}$ of each pair of mutations in the reference sequence is computed and compared to the sum $\Delta E_i + \Delta E_j$ of the effects of the two single-site mutations, individually inserted into the reference sequence. For all pairs of fixed differences between *Y. pestis* and the reference sequences, the epistatic couplings $\Delta E_{ij}$ are also recorded.

For closely diverged species MSAs, the epistatic cost between each pair of homologous sequences with no more than one gap difference (but any arbitrary number of other missense mutations) is computed as well as the proportion of fixed differences between them.

When comparing epistatic cost between pairs of fixed non-synonymous differences in *rplK* to the distance between the corresponding residues in the 3D structure of the protein, the 4V6E PDB structure is used[45]. It is displayed using PyMOL[46].

**Effective proportion of residues coupled to an amino-acid site**. DCA models are based on a matrix $J$ of pairwise epistatic couplings between residues in a sequence. The Inverse Participation Ratio (IPR) quantifies how diffuse epistatic couplings involving a residue at position $i$ are. It is computed as follows:

$$\mathrm{IPR}_i = \Sigma_{j\neq i}(J_{ij}(a_i, a_j)^2 / \Sigma_{k\neq i}J_{ik}(a_i, a_k)^2)^2 \qquad (9)$$

with $(a_1, \ldots, a_L)$ being the reference sequence.

$\mathrm{IPR}_i$ corresponds to the inverse of the effective number of sites that are epistatically coupled with a position $i$. The effective proportion of residues coupled to an amino-acid site at position $i$ in a sequence of size $L$ is derived from $\mathrm{IPR}_i$ as being $1/(\mathrm{IPR}_i \cdot L)$.

**Reporting summary**. Further information on research design is available in the Nature Research Reporting Summary linked to this article.

## Data availability

The sequence data used in this study have been deposited in the Zenodo database under accession code 5774192[47] (https://doi.org/10.5281/zenodo.5774192). The exact list of genes and Pfam domains analyzed is available at https://github.com/LucileVG/DCA_polymorphism_Ecoli/gene_domains.csv[48]. The following public databases were used: UniRef30 (2020-03), Pfam 34.0 (March 2021), Enterobase, and Mage. The reference genomes used in this study are the following: GA4805AA genome (available on NCBI[34] under BioProject accession id PRJNA218163), *Escherichia coli* K12 - chromosome ECK.1, *Escherichia coli* UMN026 - chromosome ESCUM.2, *Escherichia albertii* TW07627 - chromosome ESCAL.1, *Escherichia fergusonii* ATCC 35469T - chromosome EFER.2, *Salmonella enterica* subsp. arizonae serovar 62:z4,z23:-- RSK2980 - chromosome NC\_010067.1, *Klebsiella pneumoniae* 1162281 - WGS AFQL.1, *Atlantibacter hermanni* i 4928STDY7071316 - WGS CABGLB01.1, *Pantoea ananatis* AJ13355 - chromosome NC\_017531.1 and *Yersinia pestis* Angola - chromosome NC\_010159.1.

## Code availability

Code is available at https://github.com/LucileVG/DCA_polymorphism_Ecoli and is linked to Zenodo database under accession code 6624449,lucilevg_2022_6624449 (https://doi.org/10.5281/zenodo.6624449).

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

## Acknowledgements
We are thankful to Alaksh Choudhury for help with protein 3D structure visualization. We also wish to thank Juan Rodriguez-Rivas. Our work was partially funded by the French Agence Nationale pour la Recherche ANR GeWiEp (ANR-18-CE35-0005-01, to L.V. and O.T.), the Fondation pour la Recherche Médicale (EQU201903007848, to L.V. and O.T.), the PhD program AMX of École polytechnique and Ministére de l'Enseignement Supérieur, de la Recherche et de l'Innovation (to L.V.) and EU H2020 Research and Innovation Programme MSCA-RISE-2016 (Grant Agreement No. 734439 InferNet, to M.W.).

## Author contributions
L.V., G.C., O.T., and M.W. designed the analyses and wrote the paper. L.V. and G.C. performed the analyses. M.P. and E.R. gathered and prepared genetic sequence data.

## Competing interests
The authors declare no competing interests.
