## [Peer Review File · Nature Communications]

Deciphering polymorphism in 61,157 *Escherichia coli* genomes via epistatic sequence landscapesReviewers' Comments:

Reviewer #1:

Remarks to the Author:

The manuscript investigates the extent of genetic epistasis by contrasting the site-specific amino acid preferences predicted by a non-epistatic model with those predicted by a model with pairwise epistatic interactions. They show that the predicted site specific amino acid distributions for *E. coli* often have much less variability than would be expected based on a non-epistatic model, suggesting that epistasis is important for a substantial fraction of sites. Moreover, they show that the epistatic model can capture observed patterns of genetic variation among *E. coli* strains better than the non-epistatic model (importantly, neither of these models was trained directly on *E. coli* strains, so that this result shows that the same epistatic interactions that govern long-term protein evolution also appear to govern patterns of polymorphism within this specific species). The manuscript also investigates evolution in species that are closely related to *E. coli*, showing that site specific amino acid preferences change through the accumulation of many small epistatic interactions rather than a few large ones.

Overall the manuscript provides an innovative and highly informative method for understanding the impact of epistasis on protein evolution. Indeed the graphical depiction in Figure 3B could become a standard way of summarizing the amount of epistasis in a given system, and one could imagine comparing such bivariate scatter plots, e.g. between different types of proteins or different population genetic conditions (for example, viruses vs. bacteria vs. eukaryotes) in order to better understand differences in the strength of selection and importance of epistasis. One weakness of the manuscript is that at present the reader needs to get fairly far into the text in order to understand the basic idea of the analysis (the comparison between the entropy of the predicted site-specific amino acid distributions under the site-independent and context-dependent models). It would be helpful to the reader if the basic approach could be explained informally in the Introduction, e.g. by sketching the basic ideas as part of the current last paragraph.

I have a number of suggestions to increase the rigor of the work presented here.

(1) My understanding is that the epistatic model used here (plmDCA, Ekeberg et al. 2013) typically includes regularization and sequence reweighting that are not described in the manuscript but could potentially impact the comparison between context-dependent and context-independent models. First, by varying the regularization parameter on the pairwise couplings one can sweep the plmDCA inference from being essentially identical to the context-dependent model to being highly overfit and apparently very different from the context-dependent model. While DCA approaches have had a great deal of empirical success and appear in practice quite robust (and the polymorphism data shows that out-of-sample performance is good), this issue with the choice of regularization parameter should at least be addressed verbally (and could also be relevant to the results about the gradual accumulation of epistasis). Second, plmDCA is often used in a manner that reweights sequences in the alignment based on their similarity, but the manuscript does not state if this was done here. If reweighting is being done, it would probably be best to feed both the context-independent model and the context-dependent model the same alignment and set of weights, otherwise differences in their predictions could be due to differences in the reweighting rather than epistasis (in practice, only reweighting the context-dependent model would tend to increase the entropy of its predictions, and would therefore likely be conservative, but the issue should still be clarified).

(2) The inter-species alignments are used if they contain at least 200 sequences. These are quite small alignments, and the whole distribution of alignments should be shown so that the reader can evaluate the typical scale of the data used to train the models. In addition, trying to estimate 20 amino acid frequencies from only 200 sequences will produce downwardly biased estimates of context-independent entropy, which could in part explain why so few entropies are near the upper bound of 4.3, and in any case the issue of biased estimates should be discussed.

(3) Many readers familiar with DCA would likely be curious about the relationship between the context-independent model used here and the fields in DCA, which conceptually play a very similar role of capturing context-independent preferences, and also the extent to which the analysis could also be done formally within a Potts model / information theoretic framework. It would also be helpful to clarify the extent to which the discordance between the context-dependent and context-independent entropies can itself change depending on sequence background.

(4) The simulation procedure (line 208) needs more motivation and explanation in order to be accessible to a broad audience. In particular, these are not standard population genetic simulations and so more motivation / justification is needed in order to be convincing to evolutionary biologists (personally I find the methods crude but sufficient for the limited point for which they are employed).

Minor comments:

- Introduction. Pollock, Thiltgen, and Goldstein 2012 PNAS is also a key reference for changes in amino acid preferences over time.
- Fig 8b. What is the actual correlation between residue distance and coupling coefficient? The joint distribution looks basically independent to me.
- Line 351. "On the contrary"->"In contrast" ("On the contrary" is confusing here)
- Line 522. More background and/or a reference on the inverse participation ratio calculation would be helpful. I think it would also be helpful to emphasize in the text that this statistic is also background dependent.
- Supplemental Figures 3, 8. Figure captions should point out that the relevant simulation methods and notation are in the Supplemental Methods (the definition of lambda was hard to find).

Reviewer #2:

Remarks to the Author:

In this manuscript, the authors undertake a large-scale genomic analysis in *E. coli* and related bacteria to investigate the impact of epistasis on protein evolution. Using a Direct-Coupling Analysis, they infer widespread epistasis, largely in the form of many small effect interactions. The question is of wide interest in molecular evolution, and I am enthusiastic about the overall approach.

I do have a serious methodological concern, in that the authors do not seem to adequately address phylogenetic context. Residues at different sites can co-occur because of epistasis, but also because of shared history. It is not at all clear to me that DCA can distinguish between these two types of co-occurrence. The authors briefly address phylogenetic concerns at lines 436-442, but (a) it is not entirely clear what was done, and (b) no evidence is provided to indicate that the approach is sufficient. Covariance due to relatedness does not only affect close relatives, so reducing close relatives to a common ancestor is unlikely to mitigate these concerns. Are there simulation results that the authors can point to, to help with this concern? Or is there a way to reformulate DCA in a phylogenetic setting?

I'll keep my remaining comments brief, as the further details of the results and discussion really depend hinge on the robustness of the analysis to phylogeny. A few comments worth mentioning here though:

- The authors infer an effect of local sequence context on conservation (lines 147-192), and claim that this effect is "largely due to local epistatic couplings" (line 192). While local epistasis is one possible explanation, it is not the only one. Mutational context could also be important, for example through

the creation or destruction of mutation-prone repeat sequences.

- In the introduction, the authors open with the selectionist vs. neutralist debate, and go on to claim that recent interest in epistasis has revived this debate. I don't think this is accurate. One could imagine epistasis to be important in either a selectionist or a neutralist world. I don't really see the need to frame this study in the context of the selectionist vs. neutralist debate.
- In the introduction, a bit more information on DCA would be helpful – what is the underlying model?
- Lines 99-100 – What is deemed “too close to E. coli”?
- The reporting summary has not been filled out properly. This is a paper that is entirely based on the statistical analysis of sequence data, so responding “NA” to all of the statistical questions is surely inappropriate. Moreover, in the section on code availability, the authors state that no third-party code was used. This is beside the point. The authors note in the manuscript that code is available on GitHub – this should be reflected in the data summary.

Reviewer #3:

Remarks to the Author:

The work of Vigue et al. aims to provide a novel way to quantify the factors that contribute to sequence change at different evolutionary scales. Although the concept of epistasis has been used to explain evolutionary dynamics for a long time, traditional models of sequence variability have been typically assuming independence between amino acid sites to avoid complex model construction. The situation has changed and now data availability and models like DCA allow the estimation of those contributions, making it easier to assess their importance in real data. In this study, Vigue et al. estimate epistatic couplings among an extensive number of proteins in the E. coli genome and use these parameters to explain polymorphism in a large set of E. coli sequences that are not part of the training set. They realize that the sequence context plays an important role to predict if a given site will be polymorphic or conserved in a short evolutionary scales. In contrast with independent models that use only the variability across species to define if a given site is variable or not. A key result is the fact that the context plays a restrictive role in variability, although previous work has suggested this relationship and even used it to create evolutionary models; this work shows this with a large number of sequence data across many proteins, being able to provide strong statistical support for this phenomenon. This analysis provides evidence, as a corollary, that contingency and entrenchment exist and can be quantified in molecular evolution. Finally, another important contribution of this work is the proposition that in short evolutionary scales the magnitude of couplings might not be that relevant but as time progresses, an accumulation of multiple but non-negligible couplings contribute to strong context dependent signals. This provides statistical support to various works showing that “evolutionary energies” are predictive of fitness even if the most important couplings are not in physical contact in the fold of the protein.

In my opinion, this work is exciting and very deep. It corroborates the importance of epistasis in molecular evolution but also provides means to incorporate epistasis into the analysis of extant sequences and organisms. It is rigorous and extensive and more importantly it opens the door to better ways to explore the evolution of sequences as well as the understanding of the phenotypic effects of mutations. One potential criticism could be that when comparing predictive performance it is only done with respect to an independent model as opposed to comparison of other learning models that could account for global relationships. Having said that, I don't think that that the main goal is to highlight the predictive ability but to highlight that epistasis plays a significant role in evolution.

I do have a series of questions and comments that I would like to see addressed in a new version of this manuscript. These comments are mostly centered in clarifications and suggestions to improve readability and clarity, as I don't have technical concerns or reservations on their results or claims.

General

1. The SI should contain a list of all the proteins and Pfam domains analyzed in the article. Given that this is a long list it could be a link to a file easily accessible for review.
2. It seems that the Context Dependent Entropy (CDE) provides better predictive power to discriminate conserved sites in the context of E. Coli while CDE and CIE perform similarly for large values. I wonder if for these cases the context is actually irrelevant, i.e. does the IND model and the DCA model make similar predictive mistakes or if the possible amino acid changes from the epistatic model have different distributions of the plausible amino acids. For values near 4 I assume it is the same, but for smaller values, are the amino acid options of the two models identical ? Or does the context dependent one somehow captures better the set of choices observed in real data?
3. The authors decided to leave out the sites with < 5% polymorphisms. One argument is the possibility of sequencing errors. Could the authors provide a citation or an estimate of the sequencing error rate? Could it be as high as 4%? Somehow this number seems high to me but I might be wrong.
4. The simulations to explain the discrepancy of predictions are really interesting but somehow are not clearly explained. The explanation in the SI seems ok, but it would be good to improve a little bit the explanation in the main text and its connections to random drift.
5. I found the section on the rplK gene really interesting, specially the suggestion that not only direct contacts contribute to stability and function of the protein. The authors could also mention the case of proteins designed using an approach based on evolutionary couplings where hybrids were created and only when the sum of couplings was optimized then the hybrids would become functional. This is another case where the number of contacts is minimal but a combination of multiple favorable couplings leads to functional proteins (Jiang et al. Nat. Comm. 2021).
6. I find it intellectually honest that the authors mention that some of the signals observed might not be biological but the result of artifacts of DCA modeling. It would be even more informative, if they could explain in which way signals could be artifactual, since the effect of gaps has been removed, what could be other sources of artifacts here?

Minor

1. Results, line 95. Instead of using "good/bad" sequences I suggest you use "favorable" and "disruptive" sequences
2. Remove the space between "Methods" and a subsequent parenthesis
3. Use "performance of DCA" instead of "performances of DCA"
4. I suggest rephrasing the sentence "it thus confort us in using DCA ..." with something like "these results provide support that DCA is an adequate tool to perform further studies in this work."
5. In the section "The context constrains" please be more specific on which context, maybe " The sequence context ..." or "The amino acid context constrains .."
6. Figure 5b cannot be distinguished in a B&W version. Please use larger contrast to fix this issue.
7. Although Figure 8 looks visually pleasing, somehow the protein of interest is hard to see. Is there any advantage in showing the ribosomal environment? If not, then I would focus on showing the protein of interest only or only with one of its partners.
8. Figure 8, replace "red dots" with "red spheres"

9. Replace "artefacts" with "artifacts"

10. The "Data analysis" section is really only a couple of sentences. I suggest to merge with other sections

Reply to the reviewers' reports

We thank all reviewers for their positive evaluations and constructive comments on our manuscript. We have carefully revised our manuscript considering all reviewer's suggestions, as is detailed below in our point-by-point answers (original review in blue, our replies in black), and as can be checked easily using the manuscript with tracked changes. We hope that the manuscript is now suitable for publication in its revised form.

Answer to reviewer 1:

"The manuscript investigates the extent of genetic epistasis by contrasting the site-specific amino acid preferences predicted by a non-epistatic model with those predicted by a model with pairwise epistatic interactions. They show that the predicted site specific amino acid distributions for *E. coli* often have much less variability than would be expected based on a non-epistatic model, suggesting that epistasis is important for a substantial fraction of sites. Moreover, they show that the epistatic model can capture observed patterns of genetic variation among *E. coli* strains better than the non-epistatic model (importantly, neither of these models was trained directly on *E. coli* strains, so that this result shows that the same epistatic interactions that govern long-term protein evolution also appear to govern patterns of polymorphism within this specific species). The manuscript also investigates evolution in species that are closely related to *E. coli*, showing that site specific amino acid preferences change through the accumulation of many small epistatic interactions rather than a few large ones.

Overall the manuscript provides an innovative and highly informative method for understanding the impact of epistasis on protein evolution. Indeed the graphical depiction in Figure 3B could become a standard way of summarizing the amount of epistasis in a given system, and one could image comparing such bivariate scatter plots, e.g. between different types of proteins or different population genetic conditions (for example, viruses vs. bacteria vs. eukaryotes) in order to better understand differences in the strength of selection and importance of epistasis. One weakness of the manuscript is that at present the reader needs to get fairly far into the text in order to understand the basic idea of the analysis (the comparison between the entropy of the predicted site-specific amino acid distributions under the site-independent and context-dependent models). It would be helpful to the reader if the basic approach could be explained informally in the Introduction, e.g. by sketching the basic ideas as part of the current last paragraph."

We thank the reviewer for their careful reading of the manuscript and their valuable feedback. The writing of the introduction was a concern shared with the two other reviewers. We have revised the manuscript in order to better introduce our overall approach. In particular, we took into account Reviewer 1's suggestion that the basic ideas of the manuscript should be sketched in the last paragraph and we have modified the last paragraph of the introduction accordingly.

“(1) My understanding is that the epistatic model used here (plmDCA, Ekeberg et al. 2013) typically includes regularization and sequence reweighting that are not described in the manuscript but could potentially impact the comparison between context-dependent and context-independent models. First, by varying the regularization parameter on the pairwise couplings one can sweep the plmDCA inference from being essentially identical to the context-dependent model to being highly overfit and apparently very different from the context-dependent model. While DCA approaches have had a great deal of empirical success and appear in practice quite robust (and the polymorphism data shows that out-of-sample performance is good), this issue with the choice of regularization parameter should at least be addressed verbally (and could also be relevant to the results about the gradual accumulation of epistasis).

Several concerns are raised regarding the effect of regularization and sequence reweighting on the comparison between the context-dependent and the context-independent models. We used plmDCA with standard settings: $\theta=0.2$ (for reweighting) and $\lambda_J=0.01$, $\lambda_h=0.01$ (for L2 regularization). We revised the manuscript (section 4.4 in Materials and Methods) to mention these parameters.

We agree with the reviewer that finding the optimal regularization parameters is challenging (e.g. <https://journals.aps.org/pre/abstract/10.1103/PhysRevE.90.012132> or <https://arxiv.org/pdf/2112.01292v1.pdf>) and varying the regularization parameters can impact the DCA inference parameters (h, J). However, while we did not explore different plmDCA regularization settings, we think this has only a marginal impact on our conclusions. Indeed, we use differences of energies or entropies to evaluate polymorphisms. Both quantities are gauge independent and only weakly depend on the regularization parameters or even the inference methods.

Second, plmDCA is often used in a manner that reweights sequences in the alignment based on their similarity, but the manuscript does not state if this was done here. If reweighting is being done, it would probably be best to feed both the context-independent model and the context-dependent model the same alignment and set of weights, otherwise differences in their predictions could be due to differences in the reweighting rather than epistasis (in practice, only reweighting the context-dependent model would tend to increase the entropy of its predictions, and would therefore likely be conservative, but the issue should still be clarified).”

We thank the reviewer for bringing up this point. Indeed, in the first manuscript version we submitted, reweighting was performed only for DCA but not for IND model. We agree with the reviewer's comment on the need to train both DCA and IND models with the same alignments and set of weights, and we have updated our analyses accordingly. As expected, the results we obtained are very similar both in terms of individual amino-acid frequencies in the MSA (Pearson correlation coefficient

of 0.9832) and context-independent entropy values (Pearson correlation coefficient of 0.9816), the latter being slightly increased by reweighting (cf. figure).

“(2) The inter-species alignments are used if they contain at least 200 sequences. These are quite small alignments, and the whole distribution of alignments should be shown so that the reader can evaluate the typical scale of the data used to train the models. In addition, trying to estimate 20 amino acid frequencies from only 200 sequences will produce downwardly biased estimates of context-independent entropy, which could in part explain why so few entropies are near the upper bound of 4.3, and in any case the issue of biased estimates should be discussed.”

We agree with the reviewer that a 200 sequence threshold might be quite low. We chose it in a preliminary study where we generated full gene MSAs by querying the reference amino-acid sequence against UniRef30 2020-03 using HHblits. This quality filter allowed us to get rid of particularly small MSAs.

However, for protein domains, it is far below the typical size of a Pfam MSA (cf. new Supplementary Figure 9), only 5% of the Pfam MSAs have less than 913 sequences, while more than 60% contain >10 000 sequences. Thus, we do not believe our results to be biased by IND or DCA models trained on too small MSAs.

“(3) Many readers familiar with DCA would likely be curious about the relationship between the context-independent model used here and the fields in DCA, which conceptually play a very similar role of capturing context-independent preferences, and also the extent to which the analysis could also be done formally within a Potts model / information theoretic framework. It would also be helpful to clarify the extent to which the discordance between the context-dependent and context-independent entropies can itself change depending on sequence background.”

Due to the gauge (or reparameterization) invariance of the DCA model, taking the field parameters alone is somewhat arbitrary. It is possible to choose the gauge such that all fields are equal to those of the IND model, or to choose them to be zero, and to modify the couplings accordingly, *i.e.* without changing the DCA model. We therefore chose entropies for comparison - they are actually an information-theoretic framework, which does not depend on the gauge choice.

We also investigated how context-dependent and context-independent entropy values were evolving with sequence divergence. However, to start observing a clear signal we have to study species far more distant than the *E. coli-Y. pestis* pair. For this reason, we considered that the question was out of scope for the present study. However, we agree with the reviewer that it is an interesting point to study and might represent the basis for a future work.

“(4) The simulation procedure (line 208) needs more motivation and explanation in order be accessible to a broad audience. In particular, these are not standard population genetic simulations and so more motivation / justification is needed in order to be convincing to

evolutionary biologists (personally I find the methods crude but sufficient for the limited point for which they are employed).”

We have taken into account the request shared by reviewers 1 and 3 to better motivate and explain the use of simulations (cf. Results 2.4) and make the procedure more easily accessible (cf. Materials and Methods 4.8). The objective of these simulations is to get a rough estimate of the neutral 1-SNP non-synonymous diversity that can segregate in *E. coli* natural isolates. The main motivation behind simulations was the observation of the absence of many 1-SNP synonymous mutations across the >60,000 *E. coli* strains, a pattern consistent with genetic drift limiting the amount of neutral diversity able to segregate in a finite population.

“Minor comments:

- Introduction. Pollock, Thiltgen, and Goldstein 2012 PNAS is also a key reference for changes in amino acid preferences over time.

- Fig 8b. What is the actual correlation between residue distance and coupling coefficient? The joint distribution looks basically independent to me.

- Line 351. "On the contrary"->"In contrast" ("On the contrary" is confusing here)

- Line 522. More background and/or a reference on the inverse participation ratio calculation would be helpful. I think it would also be helpful to emphasize in the text that this statistic is also background dependent.

- Supplemental Figures 3, 8. Figure captions should point out that the relevant simulation methods and notation are in the Supplemental Methods (the definition of lambda was hard to find).”

We thank the reviewer for the minor comments. We agreed with them and modified the manuscript accordingly. In particular, we added Pollock, Thiltgen, and Goldstein 2012 PNAS to the references for changes in amino acid preferences over time in the introduction; we changed "On the contrary" to "In contrast" on line 351; we added background about IPR in section 2.6 and pointed out to the Methods section 4.8 in the caption of Supplemental Figures 3 and 8. Regarding the question of the actual correlation between residue distance and coupling coefficient, while there is only weak correlation between the two, a strong epistatic coupling is likely to be a contact (residue distance < 10Å).

Answer to reviewer 2:

“In this manuscript, the authors undertake a large-scale genomic analysis in *E. coli* and related bacteria to investigate the impact of epistasis on protein evolution. Using a Direct-Coupling Analysis, they infer widespread epistasis, largely in the form of many small effect interactions. The question is of wide interest in molecular evolution, and I am enthusiastic about the overall approach.

I do have a serious methodological concern, in that the authors do not seem to adequately address phylogenetic context. Residues at different sites can co-occur because of epistasis, but also because of shared history. It is not at all clear to me that DCA can distinguish between these two types of co-occurrence. The authors briefly address phylogenetic concerns at lines 436-442, but (a) it is not entirely clear what was done, and (b) no evidence is provided to indicate that the approach is sufficient. Covariance due to relatedness does not only affect close relatives, so reducing close relatives to a common ancestor is unlikely to mitigate these concerns. Are there simulation results that the authors can point to, to help with this concern? Or is there a way to reformulate DCA in a phylogenetic setting?”

We thank the reviewer for their thorough reading of our work. One of the main issues raised by the reviewer is the influence of phylogeny on DCA inference. In our study, we show that a large proportion of amino-acid sites that are variable across distant species remain conserved in *E. coli*. This absence of any amino-acid change at a particular site across more than 60,000 strains is a signature of purifying selection and not of phylogeny. If most of the DCA couplings were phylogenetic, there would be no added benefit of local sequence context on explaining polymorphism. In order to address more quantitatively the reviewer’s concerns, we have investigated the influence of phylogenetic couplings on our results (cf. Supplementary Results and Supplementary Methods). We followed the methodology developed by Rodriguez Horta and Weigt (Edwin Rodriguez Horta and Martin Weigt, PLOS Computational Biology, 2021) by generating randomized MSAs that share the same phylogenies and conservation profiles as the original MSAs, but all signatures of functional or structural coevolution are removed. This allowed us to train DCA models that can only capture phylogeny-induced correlations but no true epistatic couplings between sites. We show that a DCA model trained on such a randomized MSA performs worse than an IND model in predicting conserved and polymorphic sites (Supplementary Figure 10b). In comparison, a DCA model trained on the original MSA achieves better performances than the IND model, meaning that most of the couplings it captures should be biologically meaningful.

“- The authors infer an effect of local sequence context on conservation (lines 147-192), and claim that this effect is “largely due to local epistatic couplings” (line 192). While local epistasis is one possible explanation, it is not the only one. Mutational context could also be important, for example through the creation or destruction of mutation-prone repeat sequences.”

We do not agree with this comment. It is true that local sequences extending one or two bases away around a focal base may have an impact on the focal base mutation rate as well as 30 bp

secondary structure in the local RNA/DNA sequence. We however do not think that the resulting fluctuations in mutation rate should lead to a consistent coupling in protein sequence evolution because:

1. Apart from some possible very rare sequences (mostly described so far to generate frameshifts) these local fluctuations in mutation rates are small and mostly visible in mutator background (Couce *et al.*, PNAS, 2017). Given the per base mutation rate of $1e-10$ and the amplitude of the effect found (upper range 10-fold) these couplings are much less likely to leave a signature compared to the action of natural selection.
2. Moreover, these effects are volatile as they are affected by synonymous mutations that accumulate quite fast and change subsequently the local context and mutation rates. Given the time scale we use in the MSA that relies on protein sequences (and not nucleotide sequences) with quite some divergence, couplings are unlikely to emerge through that process.

“- In the introduction, the authors open with the selectionist vs. neutralist debate, and go on to claim that recent interest in epistasis has revived this debate. I don't think this is accurate. One could imagine epistasis to be important in either a selectionist or a neutralist world. I don't really see the need to frame this study in the context of the selectionist vs. neutralist debate.

- In the introduction, a bit more information on DCA would be helpful – what is the underlying model?”

The writing of the introduction is a concern that Reviewer 2 shares with both Reviewer 1 and Reviewer 3. We have addressed this concern by rewriting part of the introduction. In particular, we have taken into account Reviewer's 2 comments on the lack of background on DCA by providing more nontechnical background on DCA and how this can be used to characterize sequence variability and context-dependence of mutations. Reviewer 2 also questions the relevance of framing our study in the context of the selectionist vs. neutralist debate. We agree with the reviewer that epistasis might indeed be integrated in both a neutralist or a selectionist framework. However, the historical papers at the core of the neutral-vs-selectionist debate often implicitly base their arguments on independent-site models of protein evolution. For instance, *Non-Darwinian Evolution* (King and Jukes, 1969) argumentation relies on the study of conserved and variable sites of proteins such as Cytochrome c across distant species. We are not the first ones to notice that taking epistasis into account might lead to re-interpret the patterns of variability that are observed between homologues of the same protein and thus has strong implications on how the neutral-vs-selectionist debate is shaped (see for instance Breen et al. 2012).

“- Lines 99-100 – What is deemed “too close to *E. coli*”?”

In order to take the reviewer's comment into account, we clarified the meaning of “too close to *E. coli*”. It corresponds to sequences with more than 90% identity with the reference.

“- The reporting summary has not been filled out properly. This is a paper that is entirely based on the statistical analysis of sequence data, so responding “NA” to all of the statistical questions is surely inappropriate. Moreover, in the section on code availability, the authors state that no third-party code was used. This is beside the point. The authors note in the manuscript that code is available on GitHub – this should be reflected in the data summary.”

We have updated the reporting summary. We added the link to the GitHub code of third-party libraries we used for our analyses. Regarding statistical analyses, we filled the reporting summary following editorial recommendations during the submission of an earlier article to Nature Communications. Our understanding is that this section is dedicated to the statistical analyses of experimental data. Given that our work does not involve experiments, we consider that the questions do not apply to our situation.

Answer to reviewer 3:

“The work of Vigue et al. aims to provide a novel way to quantify the factors that contribute to sequence change at different evolutionary scales. Although the concept of epistasis has been used to explain evolutionary dynamics for a long time, traditional models of sequence variability have been typically assuming independence between amino acid sites to avoid complex model construction. The situation has changed and now data availability and models like DCA allow the estimation of those contributions, making it easier to assess their importance in real data. In this study, Vigue et al. estimate epistatic couplings among an extensive number of proteins in the E. coli genome and use these parameters to explain polymorphism in a large set of E. coli sequences that are not part of the training set. They realize that the sequence context plays an important role to predict if a given site will be polymorphic or conserved in a short evolutionary scales. In contrast with independent models that use only the variability across species to define if a given site is variable or not. A key result is the fact that the context plays a restrictive role in variability, although previous work has suggested this relationship and even used it to create evolutionary models; this work shows this with a large number of sequence data across many proteins, being able to provide strong statistical support for this phenomenon. This analysis provides evidence, as a corollary, that contingency and entrenchment exist and can be quantified in molecular evolution. Finally, another important contribution of this work is the proposition that in short evolutionary scales the magnitude of couplings might not be that relevant but as time progresses, an accumulation of multiple but non-negligible couplings contribute to strong context dependent signals. This provides statistical support to various works showing that “evolutionary energies” are predictive of fitness even if the most important couplings are not in physical contact in the fold of the protein.

In my opinion, this work is exciting and very deep. It corroborates the importance of epistasis in molecular evolution but also provides means to incorporate epistasis into the analysis of extant sequences and organisms. It is rigorous and extensive and more importantly it opens the door to better ways to explore the evolution of sequences as well as the understanding of the phenotypic effects of mutations. One potential criticism could be that when comparing predictive

performance it is only done with respect to an independent model as opposed to comparison of other learning models that could account for global relationships. Having said that, I don't think that the main goal is to highlight the predictive ability but to highlight that epistasis plays a significant role in evolution."

We thank the reviewer for their thorough analysis of our work and their useful comments. Concerning the choice of DCA compared to other learning models that could account for global relationships, we have revised the introduction of the manuscript in order to better present DCA and its pertinence to study epistasis. Other learning models (such as DeepSequence <https://www.nature.com/articles/s41592-018-0138-4> or GEMME <https://academic.oup.com/mbe/article/36/11/2604/5548199?login=true>) can likely also achieve good predictions, but, to our knowledge, none of them was developed to predict polymorphism, but to predict the mutational landscape. Also, most of them are less interpretable in terms of epistatic interactions, which are the direct model parameters in DCA. Our main motivation is to quantify the role of epistasis in constraining protein evolution, this is the reason why we compare DCA to a non-epistatic model of amino-acid sequences. However, we agree with the reviewer that if our main concern had been to assess the predictive performances of DCA in itself, we would need to compare it to more elaborate learning models.

"1. The SI should contain a list of all the proteins and Pfam domains analyzed in the article. Given that this is a long list it could be a link to a file easily accessible for review."

We now provide a file with all proteins and Pfam domains studied in our manuscript. The file is accessible on the Github repository where the code is stored (gene_domains.csv). This is also mentioned in the main manuscript.

"2. It seems that the Context Dependent Entropy (CDE) provides better predictive power to discriminate conserved sites in the context of E. Coli while CDE and CIE perform similarly for large values. I wonder if for these cases the context is actually irrelevant, i.e. does the IND model and the DCA model make similar predictive mistakes or if the possible amino acid changes from the epistatic model have different distributions of the plausible amino acids. For values near 4 I assume it is the same, but for smaller values, are the amino acid options of the two models identical ? Or does the context dependent one somehow captures better the set of choices observed in real data?"

We thank the reviewer for this interesting question. We chose to investigate whether some epistatic couplings might also constrain variable sites by comparing the amino-acid ranks of major and minor alleles observed at polymorphic sites predicted by DCA and IND. If the overall distributions have the same shape (major allele peaking at first rank and minor allele at second rank, cf. Fig 2b and newly added 2c), DCA clearly outperforms IND in predicting consensus allele and to a lesser extent minor allele. It follows that the reduction in entropy observed on some variable sites when context is imposed (positive information gain) reflects relevant epistatic constraints acting on these sites.

“3. The authors decided to leave out the sites with < 5% polymorphisms. One argument is the possibility of sequencing errors. Could the authors provide a citation or an estimate of the sequencing error rate? Could it be as high as 4%? Somehow this number seems high to me but I might be wrong.”

We agree with the reviewer that the sequencing error rate is not expected to reach 5%. However, the main motivation behind the 5%-threshold is to get rid of deleterious variants (that might affect conserved regions so lead to a wrong classification of conserved regions into variable ones). Selection is weakly acting on variants segregating at low frequency so we expect to have a lot of deleterious variants among the ones at frequencies below 5%. We have clarified this point in the manuscript.

“4. The simulations to explain the discrepancy of predictions are really interesting but somehow are not clearly explained. The explanation in the SI seems ok, but it would be good to improve a little bit the explanation in the main text and its connections to random drift.”

Reviewer 3 shares the concern with Reviewer 1 about the need to better explain the simulations. We have reformulated the presentation of these simulations (cf. Results 2.4) and we have made the procedure more easily accessible (cf. Materials and Methods 4.8). The objective of these simulations is to get a rough estimate of the neutral 1-SNP non-synonymous diversity that can segregate in *E. coli* natural isolates. The main motivation behind simulations was the observation of the absence of many 1-SNP synonymous mutations across the >60,000 *E. coli* strains, a pattern consistent with genetic drift limiting the amount of neutral diversity able to segregate in a finite population.

“5. I found the section on the rplK gene really interesting, specially the suggestion that not only direct contacts contribute to stability and function of the protein. The authors could also mention the case of proteins designed using an approach based on evolutionary couplings where hybrids were created and only when the sum of couplings was optimized then the hybrids would become functional. This is another case where the number of contacts is minimal but a combination of multiple favorable couplings leads to functional proteins (Jiang et al. Nat. Comm. 2021).”

We thank the reviewer for the suggestion and updated the rplK section accordingly.

“6. I find it intellectually honest that the authors mention that some of the signals observed might not be biological but the result of artifacts of DCA modeling. It would be even more informative, if they could explain in which way signals could be artifactual, since the effect of gaps has been removed, what could be other sources of artifacts here?”

As discussed with Reviewer 2, in addition to capturing biologically relevant couplings, DCA may also infer some phylogenetic correlations. This comes from the fact that one of DCA's main assumptions is that amino-acid sequences observed in nature represent an (up to simple

reweighting) independent sample from an unknown probability distribution. This hypothesis is violated by the phylogenetic relationships between members of the same protein family. However, we believe that phylogenetic correlations cannot explain the patterns described in this manuscript. Indeed, we show that a large proportion of amino-acid sites that are variable across distant species remain conserved in *E. coli*. This absence of any amino-acid change at a particular site across more than 60,000 strains is a signature of purifying selection not phylogeny. If most of the couplings DCA captures were phylogenetic, there could not be any added benefit of local sequence context on explaining polymorphism. To control for the influence of phylogeny on our results, we followed the methodology developed by Rodriguez Horta and Weigt (Edwin Rodriguez Horta and Martin Weigt, PLOS Computational Biology, 2021) by generating randomized MSAs that share the same phylogenies and conservation profiles as the original MSAs, but all signatures of functional or structural coevolution are removed. This allowed us to train DCA models that can only capture phylogeny-induced correlations but no true epistatic couplings between sites. We show that a DCA model trained on such a randomized MSA performs worse than an IND model in predicting conserved and polymorphic sites (Supplementary Figure 10b). In comparison, a DCA model trained on the original MSA achieves better performances than the IND model, meaning that most of the couplings it captures should be biologically meaningful. We have updated the discussion to mention the influence of phylogeny over the patterns we observe.

“Minor

1. Results, line 95. Instead of using “good/bad” sequences I suggest you use “favorable” and “disruptive” sequences
2. Remove the space between “Methods” and a subsequent parenthesis
3. Use “performance of DCA” instead of “performances of DCA”
4. I suggest rephrasing the sentence “it thus confort us in using DCA ...” with something like “these results provide support that DCA is an adequate tool to perform further studies in this work.”
5. In the section “The context constrains” please be more specific on which context, maybe “The sequence context ...” or “The amino acid context constrains ..”
6. Figure 5b cannot be distinguished in a B&W version. Please use larger contrast to fix this issue.
7. Although Figure 8 looks visually pleasing, somehow the protein of interest is hard to see. Is there any advantage in showing the ribosomal environment? If not, then I would focus on showing the protein of interest only or only with one of its partners.
8. Figure 8, replace “red dots” with “red spheres”

9. Replace “artefacts” with “artifacts”

10. The “Data analysis” section is really only a couple of sentences. I suggest to merge with other sections”

We thank the reviewer for their minor comments. We have implemented all of them. Regarding the use of the terminology “good/bad”, we chose to change it into “fully functional” and “non-functional” because we consider “favorable” and “disruptive” better describe the effect of mutations than of entire sequences.

Reviewers' Comments:

Reviewer #1:

Remarks to the Author:

In this revised manuscript, the authors have included an improved overview of their results in the Introduction, added simulations to address the effects of phylogeny, and clarified / addressed a number of technical issues such as the weighting of sequences. These modifications are on the whole minor, but have improved the rigor and presentation of this work and my view the manuscript is now ready for publication.

If I had one more minor suggestion, I do think many readers would benefit from slightly more main text explanation of the influence of phylogeny on DCA and why this influence is expected to be limited in practice (e.g. the Horta and Weigt 2021 study but also Qin and Colwell 2018, which provides some good intuitions). I think it is also perhaps worth acknowledging in the main text that phylogenetic signal alone would result in a decrease in entropy under DCA relative to IND, but not to the extent observed in the real data (as shown in supplemental figure 10A).

Typos:

- pg 2 "DCA epistatic models constantly perform better than" -> "DCA epistatic models consistently perform better than"
- pg 18 "and more than 95". Second half of sentence is missing.
- SI pg 13 "These findings are in line the conclusion in [44]". Missing "with"

Reviewer #3:

Remarks to the Author:

I have revised the rebuttal letter and found all of my questions answered in a satisfactory way. I have also revised the answers to the other reviewers and the changes and responses seem also appropriate, including those responses where the answer to the reviewer did not agree with the suggestion of the referee. I think this version of the manuscript is clearer, with more explanations for non-specialists and providing additional support for several claims.

Reply to the reviewers' reports

We thank the reviewers for thoroughly evaluating the revisions we have made to our manuscript and considering the answers we have given to their comments in our letter. We have paid attention to their evaluations and revised our manuscript accordingly, as detailed below (original review in blue, our replies in black).

Answer to reviewer 1:

"In this revised manuscript, the authors have included an improved overview of their results in the Introduction, added simulations to address the effects of phylogeny, and clarified / addressed a number of technical issues such as the weighting of sequences. These modifications are on the whole minor, but have improved the rigor and presentation of this work and my view the manuscript is now ready for publication."

We thank the reviewer for the careful assessment of the changes we have made to the manuscript and the overall positive evaluation of our study.

"If I had one more minor suggestion, I do think many readers would benefit from slightly more main text explanation of the influence of phylogeny on DCA and why this influence is expected to be limited in practice (e.g. the Horta and Weigt 2021 study but also Qin and Colwell 2018, which provides some good intuitions). I think it is also perhaps worth acknowledging in the main text that phylogenetic signal alone would result in a decrease in entropy under DCA relative to IND, but not to the extent observed in the real data (as shown in supplemental figure 10A)."

We agree with the reviewer's suggestion to detail more the explanation of the influence of phylogeny on DCA in the main text and to refer to Qin and Cowell's work. We have chosen to extend the analysis of this specific point in the discussion section of the manuscript.

"Typos:

- pg 2 "DCA epistatic models constantly perform better than" -> "DCA epistatic models consistently perform better than"
- pg 18 "and more than 95". Second half of sentence is missing.
- SI pg 13 "These findings are in line the conclusion in [44]". Missing "with"

We thank the reviewer for bringing these typos to our attention. We have corrected them.

Answer to reviewer 3:

"I have revised the rebuttal letter and found all of my questions answered in a satisfactory way. I have also revised the answers to the other reviewers and the changes and responses seem also appropriate, including those responses where the answer to the reviewer did not agree with the suggestion of the referee. I think this version of the manuscript is clearer, with more explanations for non-specialists and providing additional support for several claims."

We thank the reviewer for the positive evaluation of our work and for thoroughly reviewing not only our answer to their own comments but also the answers we have given to reviewer 1 and reviewer 2.